# DIFFERENTIAL FINE-TUNING LARGE LANGUAGE MODELS TOWARDS BETTER DIVERSE REASONING ABILITIES

**Xiaosong Yuan**[1,2,3†], **Chen Shen**[3‡], **Shaotian Yan**[3], **Kaiyuan Liu**[3,4], **Xiaofeng Zhang**[5]
**Sinan Fan**[4], **Liang Xie**[6], **Wenxiao Wang**[4], **Renchu Guan**[1,2], **Ying Wang**[1,2*], **Jieping Ye**[3]
[1]College of Computer Science and Technology, Jilin University
[2]Key Laboratory of Symbolic Computation and Knowledge Engineering, MoE, Jilin University
[3]Alibaba Cloud Computing
[4]College of Computer Science and Technology, Zhejiang University
[5]School of Automation and Intelligent Sensing, Shanghai Jiao Tong University
[6] College of Computer Science and Technology, Zhejiang University of Technology
yuanxs19@mails.jlu.edu.cn, zjushenchen@gmail.com

## ABSTRACT

Reasoning abilities of large language models (LLMs) require explicit derivations compared to general question-answering, supervised fine-tuning (SFT) can empower multiple reasoning abilities in LLMs via learning from various datasets. However, neither training the datasets jointly (mix-up) nor continually can maintain the performance of single-dataset SFT, sometimes better while sometimes even worse, illustrating vanilla SFT can not only facilitate reasoning abilities but also introduce conflicts. In this paper, we propose a novel framework to mitigate the conflicts and preserve benefits among different reasoning tasks, and even surpass each task's single dataset SFT performance. We start by exploring the differences between reasoning fine-tuned and base LLMs by analyzing their parameter variations during model inference, and we discover that each reasoning capability has exclusive parameters that benefit it more evidently than others. In contrast, the overlapped parameters of tasks can bring benefits or conflicts. Inspired by the findings, we propose to update the exclusive and overlapped parameters according to specific reasoning task combinations differentially, thereby avoiding unnecessary conflicts while maintaining benefits. Consistent improvements in mix-up and continual SFT experiments demonstrate that the proposed SFT strategy can achieve better performance on various LLMs (Llama3-8B, Mistral-7B, and Qwen2.5-14B) and diverse reasoning tasks with fewer conflicts, showing the superiority and generality of our analysis findings and the proposed approach.

## 1 INTRODUCTION

Large language models (LLMs) have emerged various reasoning abilities (Achiam et al., 2023; Dubey et al., 2024; Yang et al., 2024), such as math problem-solving (Yue et al., 2024), coding (Guo et al., 2024), logical inference (Pan et al., 2023), and commonsense reasoning (Zhao et al., 2024). In contrast to the general conversation, reasoning tasks often require models to perform higher-order cognitive processes such as analysis, deduction, and problem-solving. Supervised fine-tuning (SFT) on distinct labeled datasets can facilitate such proficiencies (Dong et al., 2023; Zhang et al., 2024; Chen et al., 2024; Lu et al., 2024), enabling LLMs with versatile reasoning capabilities. Although vanilla SFT on different reasoning data can strengthen LLMs' certain capabilities in some curated combinations (Dong et al., 2023), it tends to underperform on a single dataset, revealing that mutual enhancement and conflict may coexist across reasoning tasks. Prior works have explored the destructive interference of varied tasks (Wang et al., 2024; Hui et al., 2024b; Panda et al., 2024; Gong

---

† Work done during an internship at Alibaba Cloud Computing    ‡ Project lead
* Corresponding author

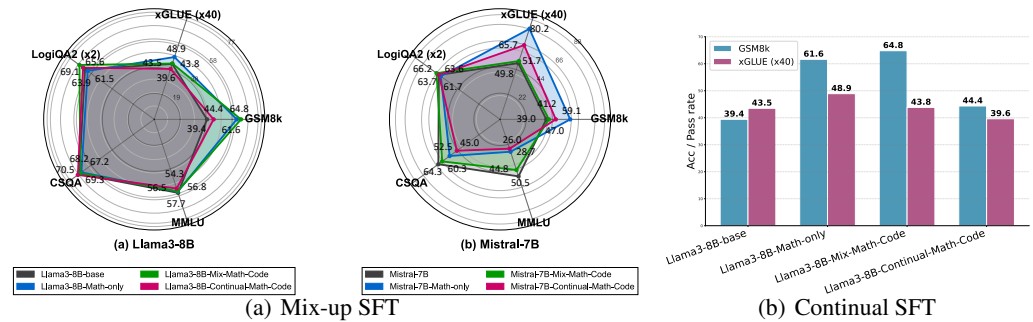

Figure 1: Performance of various math-related SFT models on 5 benchmarks with Llama3-8B and Mistral-7B, scores are increasingly ranked from the center to the circle. (We multiply the pass rate of xGLUE by 40 and the accuracy of LogiQA2 by 2 to align with others for better visualization.)

et al., 2024), they focused on the conflict of general abilities rather than reasoning and believed that all of interactions were harmful to involved tasks.

In the investigation, we conduct comprehensive SFT experiments with different LLMs on types of reasoning data to discover the relationships among various reasoning proficiencies. As shown in Figure 1(a), some combinations, like `Mix-Math-Code` of Llama3-8B, obtain significant improvements in math (measured by GSM8k) compared to `Math-only`, while it underperforms on other tasks like code (measured by xGLUE), which is more clear in Figure 1(b). On the other hand, continual learning results through `Continual-Math-Code` exhibit severe negative interference. However, an intriguing difference emerges in Mistral-7B (Jiang et al., 2023), suggesting complex dynamics in distinct LLMs. These tendencies are also exhibited similarly in combinations among more reasoning tasks, while they perform distinctly in different LLMs. Such phenomena imply benefits and conflicts between distinct reasoning capabilities that may be ubiquitous. More detailed experiments and analysis are introduced in Section 3.

Previous efforts have been made in parameter-variation SFT to mitigate conflicts among various LLM abilities. Dong et al. (2023) designed a dual-stage mixed SFT strategy to endow LLMs with math and other capabilities. HFT (Hui et al., 2024b) updated the half LLM parameters randomly in continual SFT to alleviate catastrophic forgetting. LoTA (Panda et al., 2024) employed task vector extraction and sparse adaptation to minimize interference among tasks. Nonetheless, the complete picture of task relationships is neglected, encompassing beneficial, contradictory, and neutral. In this paper, we investigate the mutual benefits and conflicts of reasoning capabilities in the SFT process.

To determine what benefits and conflicts exist and what causes those, we explore the intrinsic weights of distinct fine-tuned LLMs. Concretely, we present a novel approach to identify the individual parameter sensitivity via sampled data inference on corresponding LLMs, thereby locating necessary weights for specific abilities. After that, we design a suit of *Differential* SFT (DiFT) strategies to get better versatile reasoning abilities: for mix-up SFT, we merely fine-tune the parameters union of critical weights for involved tasks, to obtain target reasoning abilities while making less disturb to others; as for continual SFT, we freeze the parameters in difference set of the former and current tasks, to reserve historic proficiencies and learn new ones by remaining parameters.

We employ base instead of instruct LLMs for analysis and validation, as instruct LLMs have been through massive post-training, making it hard to measure their inner benefits/conflicts. Additionally, *our fine-tuned LLMs with fewer data beat instruct LLMs on some tasks (e.g., logic and commonsense in Section 5.4), highlighting that our research can provide insights for reasoning-oriented fine-tuning(regardless of base or instruct models).* We conduct extensive experiments with pilot LLMs on several reasoning tasks, and results show that the proposed DiFT can improve all LLMs in various reasoning combinations, where mix-up SFT can approach the single dataset SFT, and continual SFT can reduce more historical performance losses, demonstrating that our analysis is valid and DiFT can mitigate reasoning conflicts and keep mutual benefits. Our contributions are as follows:

- We investigate in comprehensive SFT experiments on single (vanilla), mix-up, and continual reasoning datasets with different LLMs, showing that mutual benefits and conflicts exist among distinct reasoning tasks commonly.

- By analyzing the parameter variations during inference between various fine-tuned and base models, we discover that some parameters are vital to specific reasoning tasks, i.e., each reasoning capability corresponds to certain parts of parameters.

- Based on the analysis and findings, we propose a novel fine-grained SFT strategy to preserve enhancement and mitigate the conflicts by selectively updating those reasoning-relevant parameters of LLMs.

- We conduct extensive experiments on different LLMs with the proposed DiFT, and empirical results across distinct reasoning tasks are in line with our analysis, validating the effectiveness of the proposed approach.

## 2  RELATED WORK

SFT has been demonstrated as a productive post-training paradigm for improving models' various capabilities (Minaee et al., 2024), including math (Yu et al., 2023), code (Guo et al., 2024), commonsense (Bian et al., 2024), logic (Chen et al., 2023), and instruction following (Lou et al., 2023). There are task conflicts in LLMs (Luo et al., 2023), and numerous SFT variants emerged in the era of LLMs from data selection and optimization. Task Vector (Ilharco et al., 2022) considered the fine-tuned and pre-trained parameter variations as the task-related weights and conducted addition and negation to modify or combine different tasks. Rudman et al. (2023) discovered that outlier dimensions could encode crucial task-specific knowledge and that the value of a representation in a single outlier dimension drives downstream model decisions. Zhang et al. (2023) proposed parameter optimization trajectory and learned to uncover intrinsic task-specific subspace by exploiting the dynamics of fine-tuning a given task. Nonetheless, these works failed to connect the specific parameters and tasks.

DMT (Dong et al., 2023) designed dual-stage mixed fine-tuning to endow LLMs with math, code, and instruction abilities. These methods aim to find better data usage, ignoring the learning process. Tang et al. (2024) presented a partial linearization to fuse multi-task abilities into one model. HFT (Hui et al., 2024b) updated a random half of LLM parameters in continual fine-tuning to alleviate catastrophic forgetting. LoTA (Panda et al., 2024) employed task vector extraction and sparse adaptation to minimize interference among multiple tasks. Kong et al. (2024) introduced a gradient approximation strategy for activated parameter locating to reduce the computational complexity associated with many parameter partitions. Nevertheless, none of them analyzes the model parameters in-depth from the perspective of reasoning conflicts.

## 3  BENEFITS AND CONFLICTS ANALYSIS

We intend to validate and explore the mutual benefits and conflicts among reasoning abilities by delving into the LLMs' parameters step by step to explore the causes. First, we conduct SFT experiments on 4 datasets in 3 settings: vanilla, mix-up, and continual (detailed experimental settings in Section 5.1). As instruct-LLMs were trained on massive data, the scaling-up training may trade-off a part of conflicts in math reasoning, we take Llama3-8B-base and Mistral-7B-base, results are shown in *Table 1*. **The findings and intervening phenomena are similar in instruct-LLMs in Section 5.4**.

### 3.1  MIX-UP AND CONTINUAL REASONING SFT

In *Table 1*, vanilla SFT enhances the corresponding reasoning ability stably on both Llama and Mistral, while it may affect others: for example, the `Math-only` can degrade logic and commonsense a bit, and so do the `Logic-only` and `CSQA-only` to math. Such results suggest:

 i. *There may be a learning trade-off between distinct abilities that leads to reasoning interference.*

Interestingly, the mix-up SFT exhibits synergistic effects in both positive and negative aspects. `Mix-Math-Code` achieves great performance on both GSM8k and xGLUE compared to single variants, implying math and code may share complementary weights, evidenced by the 64.82% accuracy and 1.0956 pass rate of `Mix-Math-Code` on Llama3-8B, surpassing `Math-only`. An

Table 1: The mix-up and continual SFT results of Llama3-8B and Mistral-7B on 5 benchmarks, the ↓and ↑denote decreasing and increasing compared to the base model performance, respectively.

| Methods | Llama3-8B | | | | | Mistral-7B | | | | |
|---|---|---|---|---|---|---|---|---|---|---|
| | GSM8k | xGLUE | LogiQA2 | CSQA | MMLU | GSM8k | xGLUE | LogiQA2 | CSQA | MMLU |
| base model | 39.42 | 1.0874 | 31.93 | 69.29 | 57.66 | 38.97 | 1.2449 | 31.87 | 64.29 | 50.49 |
| | *Vanilla SFT* | | | | | | | | | |
| Math-only | 61.64↑ | 1.2228↑ | 30.73↓ | 67.24↓ | 56.76 | 59.14↑ | 2.0042↑ | 30.85↓ | 52.50↓ | 28.68↓ |
| Code-only | 26.54↓ | 1.1203↑ | 35.05↑ | 70.93↑ | 55.65 | 31.31↓ | 1.7146↑ | 28.94↓ | 58.39↓ | 43.04↓ |
| Logic-only | 30.17↓ | 0.6880↓ | 37.02↑ | 72.89↑ | 57.52 | 4.62↓ | 1.3628↑ | 31.23 | 54.55↓ | 32.47↓ |
| CSQA-only | 8.79↓ | 0.5702↓ | 29.90↓ | 79.36↑ | 28.10↓ | 1.36↓ | 2.7964↑ | 30.15↓ | 70.93↑ | 23.43↓ |
| | *Mix-up SFT* | | | | | | | | | |
| Mix-Math-Code | 64.82↑ | 1.0956 | 34.54↑ | 68.22 | 56.47 | 41.17↑ | 1.2913↑ | 33.08↑ | 60.28↓ | 44.81↓ |
| Mix-Math-Logic | 64.37↑ | 1.2092↑ | 32.32↑ | 70.52 | 55.30↓ | 57.39 | 0.8593↓ | 31.87 | 62.49 | 36.68↓ |
| Mix-Math-CSQA | 68.92↑ | 1.1342↑ | 32.32↑ | 77.31↑ | 47.46↓ | 52.77 | 2.8439↑ | 31.11 | 73.05↑ | 39.76↓ |
| Mix-Code-Logic | 52.31↑ | 1.0779 | 32.57↑ | 70.52↑ | 58.05 | 22.37↓ | 1.2342 | 31.17 | 62.00 | 43.33↓ |
| Mix-Code-CSQA | 52.39↑ | 0.8905↓ | 31.42 | 77.15↑ | 32.08↓ | 26.69↓ | 1.3969↑ | 33.46↑ | 75.02↑ | 44.97↓ |
| Mix-Logic-CSQA | 16.91↓ | 0.2150↓ | 32.44↑ | 77.40↑ | 47.14↓ | 16.60↓ | 0.9582↓ | 31.11 | 74.69↑ | 45.34↓ |
| | *Continual SFT* | | | | | | | | | |
| Continual-Math-Code | 44.35↑ | 0.9902↓ | 32.82↑ | 70.52 | 54.28↓ | 47.01↑ | 1.6431↑ | 31.81 | 44.96↓ | 25.98↓ |
| Continual-Math-Logic | 10.99↓ | 0.6433↓ | 31.30 | 67.90 | 51.53↓ | 4.62↓ | 1.0365↓ | 29.26↓ | 40.29↓ | 24.56↓ |
| Continual-Math-CSQA | 3.87↓ | 0.5494↓ | 31.36 | 78.71↑ | 47.07↓ | 1.14↓ | 3.8740↑ | 30.34↓ | 57.90↓ | 23.12↓ |

intriguing discovery is the imbalance impact: `Mix-Math-Code` improves math from 61.64% to 64.82%, while it only enhances xGLUE to 1.0956, a minor less than `Code-only`, implying:

> ii. *Benefits between different reasoning abilities are not always reciprocal, where one reasoning ability may gain more than the other.*

In contrast to mix-up SFT, continual SFT is born with catastrophic forgetting, which remains a significant challenge, making it more complex (Ramasesh et al., 2021; Luo et al., 2023). At the bottom of Table 1, we can hardly observe benefits between reasoning abilities. While `Continual-Math-Logic` achieves moderate logic performance (31.30% on Llama3-8B), it shows severe degradation in math reasoning (10.99% on GSM8k). Such results indicate that continual SFT with different reasoning data may lead to the erosion of previously acquired abilities. Additionally, the continual SFT on one task performs worse than the direct SFT on the base LLM in some settings, e.g., significant task interference in `Continual-Math-Code` (1.084 to 0.9902 with Llama3-8b). Such results indicate that there also exist reasoning conflicts besides catastrophic forgetting. Therefore, we make an assumption:

> iii. *Even catastrophic forgetting is the main issue, reasoning conflicts matter in continual SFT.*

The above findings highlight the complex interactions between different reasoning capabilities and the challenges in mitigating conflicts while preserving benefits. To address the above challenges, we start by analyzing the inner weights of LLMs with different reasoning proficiencies.

## 3.2 DELTA-SCALE ROWS

We propose a novel approach for identifying influential weights in LLMs inspired by Yu et al. (2024), and to quantify the sensitivity of the model output to changes in weights, we introduce the *delta-scale row* (DSR) score to measure the parametric sensitivity. Let $W \in \mathbb{R}^{H \times D}$ denote the weight matrix of a linear layer, with $H$ as the out-dimension and $D$ the in-dimension, $\mathcal{L}(W)$ is the loss of the current task on $W$; $W^0$ is parameters of the original model while $W^f$ is of the fine-tuned; $\Delta W_k = W_k^f - W_k^0$, $X_t$ is an input sample at token $t$, and $Y_t = X_t W^\top$. The fine-tuning process can be formalized as:

$$W^{s+1} = W^s - \eta_s g^s \implies \Delta W_k = -\int_0^T g_k(\tau)d\tau \tag{1}$$

where $g^s$ is gradient at step $s$, $\eta_s$ is learning rate, $\tau$ is integration variable, and $T$ is total steps. Hence, $|\Delta W_k|$ is large if the gradient keep projecting on $k$-th row significantly in training. We perform a second-order Taylor expansion by varying $W_k$, plug $\delta = \Delta W_k$, and obtain:

$$\Delta \mathcal{L}_k = -g_k \Delta W_k - \tfrac{1}{2} H_{kk}(\Delta W_k)^2 + o\big((\Delta W_k)^2\big) \tag{2}$$

where $H$ is the Hessian matrix of the loss ($H = \nabla_\theta^2 \mathcal{L}(W^0)$) at the starting point $W^0$, and $H_{kk}$ is its $k$-th diagonal element (curvature along the $k$-th row). Substantial loss drop thus needs a non-trivial $|\Delta W_k|$. For a set of input $X$, the output activations $Y \in \mathbb{R}^{L \times H}$ are typically computed as:

$$Y = XW^T + b \tag{3}$$

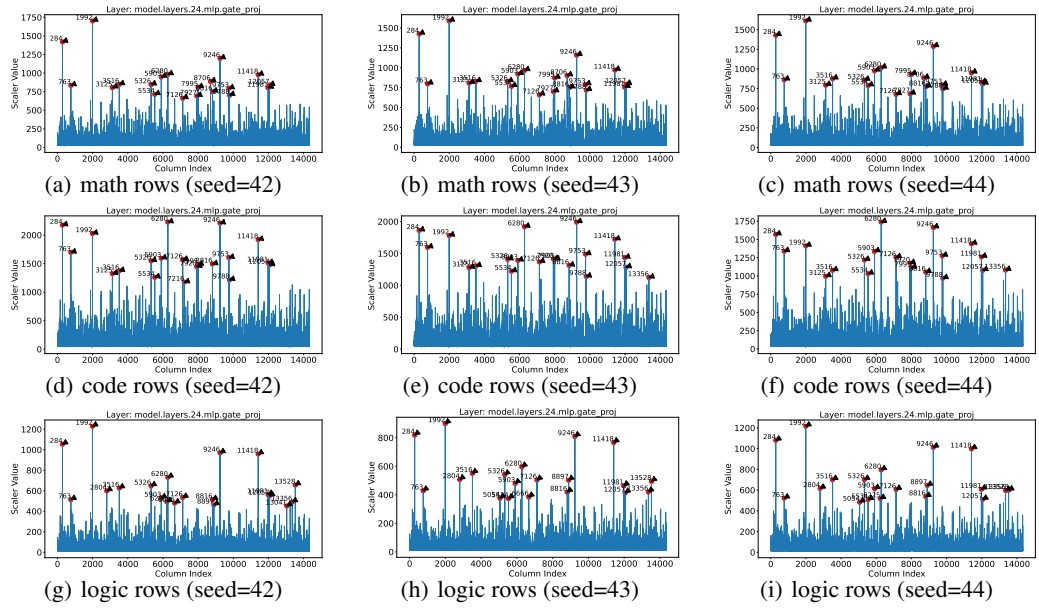

Figure 2: Distribution of delta-scale rows (DSR) for model.layer.24.mlp.gate.proj with distinct sampled data on different reasoning LLMs, we randomly sample 3 times (with seeds 42/43/44) for each task. The $x$-axis denotes the row order of the specific weight matrix, and $y$-axis is DSR values.

We analyze the difference in outputs between a base model ($M_{base}$) and a fine-tuned model ($M_{ft}$), where weights are presumed to have changed during fine-tuning. Let $Y_{base}$ and $Y_{ft}$ be the output activations of a specific layer for the same input $X$ from $M_{base}$ and $M_{ft}$, respectively. The difference in output for the $k$-th component (corresponding to the $k$-th row of $W$) for a given token $t$ is:

$$\Delta Y_t^k = Y_{ft}^k(t) - Y_{base}^k(t) \tag{4}$$

This $\Delta Y_t^k$ reflects the impact of the accumulated changes between $W_{ft}^k$ and $W_{base}^k$ on the $k$-th output feature for that token. Combining the above derivations, we can obtain:

$$\Delta Y_t^k = X_t \Delta W_k, \quad \mathbb{E}_t[\|\Delta Y_t^k\|_2^2] = \|\Delta W_k\|^2 \mathbb{E}_t[\|X_t\|^2] \tag{5}$$

Thus, $s_k = \mathbb{E}_t[\|\Delta Y_t^k\|_2^2]$, which can be computed by $s_k = \frac{1}{N}\sum_{t=1}^{N}\|\Delta Y_t^k\|_2^2$ is proportional to both $\|\Delta W_k\|^2$ and $\|X_t\|^2$, meaning that DSR score can denote the task-specific accumulated importance.

In practice, DSR scores accumulate the squared differences for each output component $k$, effectively capturing the impact of changes in the corresponding $k$-th row of the weight matrix across reasoning data. **High values in the vector of scores indicate rows of the weight matrix (and their associated output features) that exhibit greater changes in activation magnitude due to fine-tuning, suggesting these rows are influential in the processes modified or learned by the model**.

### 3.3 Fine-tuned Reasoning Model Analysis

To analyze the *DSR* scores of parameters, we perform inference with distinct fine-tuned and base LLMs on samples, ensuring that each sampled data corresponds to a tuned LLM. Concretely, we compute for each layer in the forward pass with 5 sampled groups of 50 data items (random seed 42-46) to obtain the top DSR of each task, we display some row distributions of model.layers.24.mlp.gate.proj in *Figure 2*, other weights also express similar patterns, more visualizations in Appendix F.6. The magnitude of the DSR provides a quantitative measure of specific parameters' influence, where higher values matter more. Across all sub-figures in *Figure 2*, we can observe distinct peaks in DSR parameters. These peaks indicate specific rows in the weight matrix disproportionately affect LLMs, and the rows correspond to the critical *DSR parameters* we aim to identify. Note that we only annotate the top-20 DSR parameters for better visualization, there are remarkable differences among distributions of distinct data in Figure 2.

In the distribution of the math (Figures 2(a) to 2(c)) row-wise, distinct peaks at multiple rows, e.g., 284, 1992, and 9246, suggest that specific rows in the weight exert significant influences on the LLM's reasoning on math. Interestingly, such a pattern is consistent across sampled data with different seeds, and so are the code and logic, implying stability of key DSR parameters for the task regardless of sample variations. However, the distribution of each fine-tuned reasoning LLM exhibits distinctly in Figures 2(a) to 2(g) column-wise: rows 284 and 1992 have the top-2 scores across all rows in math and logic LLMs, while the top-2 rows of the code LLM are 6280 and 9246; the logic model has influential rows of index $>13000$, but the indices of all math DSR are $<13000$. Similar phenomena exist in more LLMs as shown in Figures 11 and 12 in Appendix F.6.

We further analyze the same model (`Math-only`) with sampled data subsets and observe a more diverse DSR parameter distribution among distinct data in Appendix F.6, illustrating the parameter divergence of reasoning abilities within LLMs. After meticulous DSR analysis, we discover that: **On the one hand, rows of the parameter matrix are not sensitive to different inputs of the same reasoning task, on the other hand, different tasks demonstrate unique parameter distributions**.

## 4 METHOD

We propose *Differential* SFT (DiFT), a novel fine-tuning strategy for LLMs which can mitigate task conflicts and maintain benefits in multi-task and continual learning settings by leveraging DSR analysis. DiFT identifies task-specific DSR parameters through activation differences between base and fine-tuned LLMs, then differentially updates those crucial to current tasks while preserving the performance of others. By isolating and protecting these sensitive yet non-specialized parameters, which can be treated as a reliable proxy for task specificity (Deng et al., 2025), DiFT reduces negative transfer and forgetting. Algorithm 1 in Appendix C provides more details.

### 4.1 DELTA-SCALE ROW ANALYSIS

Given a base model $M_{base}$ and a set of reasoning-specific LLMs $\{M_{ft}^k\}_{k=0}^{K-1}$ with corresponding datasets $\{D_k\}_{k=0}^{K-1}$, DiFT analyzes how fine-tuning alters internal activations. For each task $k$, we sample $N$ inputs from $D_k$ and register forward hooks on target layers of both $M_{base}$ and $M_{ft}^k$ to capture input/output activations. For every input $x$, we compute the difference in layer outputs:

$$\Delta h^{(l)} = h_{ft}^{(l)}(x) - h_{base}^{(l)}(x) \tag{6}$$

where $h^{(l)}$ is the output of layer $l$. We accumulate the squared L2 norm of the differences across samples to obtain per-row sensitivity (Eq. 5). The DSR scores reflect how much each weight row (i.e., neuron activation) changes during SFT. We define $DSR_k$ as the set of top-$C$ highest-scoring rows per layer for task $k$, denoting parameters most critical to its reasoning behavior. According to Section 3.2, $DSR_k$ captures the "accumulated effects" of $k$. Combining with our conflicts validation, when two tasks $A$ and $B$ conflict, $S_A = DSR_A$ and $S_B = DSR_B$ are crucial parameter sets for $A$ and $B$. Naturally, the other parameters out of these sets are more likely to hurt involved tasks, so we choose to freeze those to mitigate conflicts. We also conduct experiments to demonstrate that the effectiveness of DiFT lies in causally conflict mitigation instead of regularization, and more details can be referred to Appendix D.2 and D.5. We further extend the DSR explanations and other relevant concepts in detail in Appendix D.1.

### 4.2 MIX-UP FINE-TUNING

To fine-tune a given LLM on multiple reasoning tasks simultaneously, DiFT computes the influential union set $DSR_{union}$ and merely updates these parameters:

$$DSR_{union} = \bigcup_{k=0}^{K-1} DSR_k \tag{7}$$

The LLM is then fine-tuned on the mixed data $\bigcup_{k=0}^{K-1} D_k$. Concretely, for $S_A$ and $S_B$ that are critical for tasks $A$ and $B$, respectively, what we focus on when considering A and B is to learn with $S_A$ and $S_B$ to maintain their performance as much. Although $S_A$ and $S_B$ may disturb each other, such

conflicts cannot be measured precisely and are out of this paper's scope. Therefore, we care about the major and leave the minor, i.e., reserve the base performance of $A$ and $B$. As for $1 - S_A \cup S_B$, since we observe that $A$ and $B$ have conflicts, and $S_A$ and $S_B$ should carry more weights of maintaining the performance of $A$ and $B$ (as stated in the former paragraph), so our focus on $1 - S_A \cup S_B$ is freezing them to prevent these parameters from their potential conflicts to $A$ and $B$.

### 4.3 CONTINUAL FINE-TUNING

In continual SFT, DiFT mitigates conflicts by updating only newly important DSR parameters. Inspired by the common practice in continual learning (Wang et al., 2024), at step $k$ (for $k \geq 2$), we obtain a highly sensitive parameter set to task $k$ but not identified as critical previously:

$$DSR_{diff} = DSR_k \setminus \bigcup_{j=0}^{k-1} DSR_j \tag{8}$$

Only the $DSR_{diff}$ parameters are trained on $D_k$, starting from the latest $M_{ft}^{k-1}$. Concretely, for historic task $A$, we freeze the $S_A$ to decrease the negative influence from the current task $B$ to $A$, as $S_A$ matters a lot for task $A$; for task $B$, we employ $S_B - S_A$ to improve $B$'s performance as much. By restricting updates to parameters with less impact on prior tasks, DiFT preserves historical abilities while acquiring new skills, effectively decoupling new learning from old representations. Additionally, **given the parametric nature, DiFT can also be employed jointly with data-driven methods to reduce conflicts after mitigating forgetting**, and we will discuss that in Section 5.3. Based on the DSR analysis, fine-tuning $1 - S_B$ can also be a solution, however, the empirical results in Appendix D.6 suggest that the $DSR_{diff}$ is better.

In summary, DiFT employs DSR to identify task-specific parameters with a few reasoning data, then differentially fine-tunes to mitigate conflicts while retaining benefits. It attempts to enhance diverse reasoning proficiencies of LLMs by curated adaptation where it matters most in mix-up or continual scenarios, and disturb less what already works. **We further perform a comprehensive ablation studies in Appendix F.6 to analyze the DiFT in various perspectives**.

## 5 EXPERIMENTS

To validate our findings and evaluate the proposed DiFT, we conduct comprehensive experiments on both mix-up and continual settings. We employ Qwen2.5-3B, Llama3-8B, Mistral-7B, and Qwen2.5-14B, and 4 common reasoning benchmarks. All DiFT results are based on the top-100 DSR union, all SFT training experiments come to convergence as in Appendix B.1. 3B and 14B experiments are in Appendix E, and more ablation studies for DSR choice strategies are in Appendix D.2 to D.6.

### 5.1 SETTING

**Training data** We collect and sample reasoning data to fine-tune LLMs toward distinct abilities. All the source data are widely used for task-specific training, including but not limited to MathInstruct (Yue et al., 2023), Code Bagel Hermes (Teknium, 2023), LogiCoT (Liu et al., 2023b), and CommonsenseQA (Talmor et al., 2018). More source data can be referred to Appendix B.4. We sample 20,000 for each reasoning ability and conduct SFT involving 2 tasks with DiFT each time.

**Baselines** As DiFT works in both mix-up and continual settings, we implement comparable approaches to evaluate the performance. **HFT** (Hui et al., 2024b) is a continual SFT framework, it randomly freezes half of each named parameter when fine-tuning on a new task to memorize the old knowledge. **LoTA** (Panda et al., 2024) is an advanced gradient-projection method, in each SFT round, it extracts and masks the feature vectors in the next round. Dual-stage mixed SFT (**DMT**) (Dong et al., 2023) presented a two-stage mix-up fine-tuning strategy, implemented by merging different training data. **CoBa** (Gong et al., 2024) designed a synthesized loss by calculating the relative and absolute convergence scores. The hyperparameter settings of baselines are the same as DiFT, in Appendix B.1, where we also compare PEFT and other continual methods in Appendix F.3 and F.4.

Table 2: The mix-up and continual SFT results on 4 benchmarks. The SOTA results across different strategies are marked in **bold numbers**, and the sub-optimal results are *italic numbers*. * denotes the existence of format issues, and more details about the orthogonal combination of the proposed method and the data-driven method for reducing conflicts and mitigating forgetting are in Section 5.3.

| Methods | Llama3-8B | | | | | Mistral-7B | | | | |
|---|---|---|---|---|---|---|---|---|---|---|
| | GSM8k | xGLUE | LogiQA2 | CSQA | ATA | GSM8k | xGLUE | LogiQA2 | CSQA | ATA |
| base model | 39.42 | 1.0874 | 31.93 | 69.29 | – | 38.97 | 1.2449 | 31.87 | 64.29 | – |
| *Mix-up SFT* | | | | | | | | | | |
| Mix-Math-Code | 64.82 | 1.0956 | 34.54 | 68.22 | 59.80 | 41.17 | 1.2913 | 33.08 | 60.28 | 52.87 |
| +DMT | 65.07 | **1.0851** | 32.44 | 67.52 | 59.66 | 42.13 | *1.2400* | 32.18 | 59.82 | *52.07* |
| +CoBa | *66.21* | 1.0725 | 33.15 | 68.34 | *59.91* | **43.07** | 1.1900 | 31.94 | 58.45 | 51.29 |
| +DiFT (**ours**) | **67.02** | *1.0735* | 32.63 | 68.39 | **60.35** | *42.46* | **1.3429** | 33.33 | 59.46 | **54.80** |
| Mix-Code-Logic | 52.31 | 1.0779 | 32.57 | 70.52 | 43.23 | 22.37 | 1.2342 | 31.17 | 62.00 | 46.44 |
| +DMT | 50.37 | *1.0865* | 31.93 | 69.36 | 43.12 | 26.58 | 1.2308 | *30.62* | 63.19 | 46.08 |
| +CoBa | 51.12 | 1.0811 | *32.25* | 68.67 | *43.15* | 26.05 | *1.2431* | 30.16 | 63.82 | *46.16* |
| +DiFT(**ours**) | 41.09 | **1.1359** | **33.40** | 68.55 | **45.10** | 31.69 | **1.2555** | **32.51** | 62.24 | **47.64** |
| Mix-Logic-CSQA | 16.91 | 0.2150 | 32.44 | 77.40 | 54.92 | 16.60 | 0.9582 | 31.11 | 74.69 | 52.90 |
| +DMT | 13.79 | 0.3907 | 31.68 | *78.84* | 55.26 | 18.58 | 0.7731 | *30.72* | 72.75 | 51.74 |
| +CoBa | 14.93 | 0.3868 | *32.16* | 78.05 | 55.11 | *19.42* | 0.7847 | 30.41 | 73.48 | *51.95* |
| +DiFT(**ours**) | 16.22 | 0.4592 | **32.38** | **78.95** | **55.67** | 21.68 | 0.6196 | **31.68** | **74.45** | **53.07** |
| *Continual SFT* | | | | | | | | | | |
| Continual-Math-Code | 44.35 | 0.9902 | 32.82 | 70.52 | 46.93 | 47.01 | 1.6431 | 31.81 | 44.96 | 64.58 |
| +HFT | *44.74* | *1.0362* | 33.94 | 69.69 | *48.28* | *47.72* | 1.3429 | 31.46 | 45.95 | *57.43* |
| +LoTA | 44.29 | 1.0258 | 34.45 | 68.99 | 47.79 | 47.15 | *1.3534* | 31.92 | 45.49 | 57.41 |
| +DiFT(**ours**) | **46.32** | **1.0557** | 35.86 | 70.93 | **49.55** | **49.81** | **1.6362** | 31.81 | 44.55 | **65.81** |
| Continual-Math-Logic* | 10.99 | 0.6433 | 31.30 | 67.90 | 21.15 | 4.62 | 1.0365 | 29.26 | 40.29 | 16.94 |
| +HFT | *11.06* | 0.6682 | *31.55* | 67.52 | *21.31* | 6.57 | 0.9902 | 28.48 | 43.16 | 17.53 |
| +LoTA | 10.89 | 0.6749 | **31.87** | 66.84 | **21.38** | *6.70* | 0.9803 | *28.76* | 42.51 | *17.73* |
| +DiFT(**ours**) | **11.37** | 0.6919 | 31.23 | 68.80 | 21.30 | **10.92** | 0.7107 | **29.20** | 42.92 | **20.06** |

**Evaluation**   We choose the pass rate (code) and 0-shot accuracy to evaluate the performance of LLMs, details are in Appendix B.5. As our purpose is to reserve benefits and mitigate conflicts, we emphasize the performance of involved tasks, therefore, we use the average target accuracy (ATA) to better show gains and drops of target/historic abilities compared to base LLMs, which reflects various methods better. For math and logic, we compute the (math + logic) / 2 accuracy as the ATA score, we also multiply the code pass rate by 50 to align with other metrics (non-weighted ATA in Appendix F.2). **Computing costs:** DiFT conducts merely a small-scale SFT with 1k training data and inference on a few samples, whose computing costs are lighter than task-vector and gradient-based methods. More computation cost and complexity details are in Appendix B.3 and B.2.

## 5.2   MIX-UP SFT

*Table 2* presents the mix-up SFT results, we can observe the extreme difficulty of baselines to improve the vanilla SFT. In contrast, DiFT consistently improves the ATA, i.e., the averaged target performance, and outperforms baselines across benchmarks and LLMs. Concretely, in the `Mix-Math-Code`, we know that these 2 reasoning abilities can benefit each other. In Llama3-8B, the math reasoning benefits more, so its ATA gain of DiFT is not striking, even if it beats the baselines. While Mistral-7B fails to achieve mutual benefits much with vanilla SFT, the 2 tasks gain more (from 52.87 to 54.80) with DiFT. In `Mix-Code-Logic`, DiFT on both 2 models can improve involved reasoning abilities.

However, we notice that it hurts the math of Llama3-8B and the commonsense of Mistral-7B, which results from the `Mix-Code-Logic` not considering the DSR parameters of the math reasoning. Considering math and commonsense, issues like this can be eliminated, as in Figure 7. `Mix-Logic-CSQA` is similar to `Mix-Math-Code`, albeit the vanilla SFT has mutual benefits in Llama3-8B, the proposed DiFT still can enhance their ATA performance, as for Mistral-7B, the vanilla and all baselines trade the logic ability for commonsense, DiFT maintains more LogiQA2 accuracy (31.68%) and obtains higher CSQA accuracy (74.45%), achieving diverse performance.

In multiple mix-up SFT experiments, DiFT can maintain and facilitate mutual benefits and alleviate conflicts between reasoning capabilities significantly, thereby supporting the effectiveness of the earlier DSR analysis. We also found that math and code tasks are somehow synergistic while logic and commonsense are conflicting, which is interesting. The math-code synergizing may come from the fact that the two tasks share similar computation backgrounds, providing more views for LLMs to understand the reasoning process, and such `Mix-Math-Code` tuning has been utilized in math- and code-specific LLMs training (Shao

et al., 2024; Hui et al., 2024a). In contrast, logic tasks need to obey strict and complex rules, while commonsense tasks are more about ground knowledge and simple reasoning, leading to conflicts between the two tasks (Song et al., 2024).

We also conduct more mix-up experiments involving 4 and 5 tasks to illustrate the consistency of the DiFT. More details can be referred to Appendix F.5.

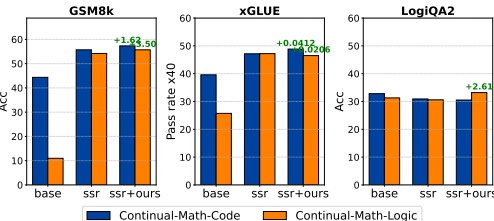

Figure 3: Joint DiFT-SSR for conflicts after alleviating forgetting.

## 5.3 CONTINUAL SFT

The bottom of *Table 2* manifests results of continual SFT, where models are fine-tuned sequentially and need to retain the previous reasoning proficiency while adapting to new ones. As mentioned in Section 3.1, reasoning benefits and conflicts exist along with catastrophic forgetting, not dominant but still important. In *Continual-Math-Code*, DiFT learns code ability better and keeps more math reasoning on both 2 LLMs, resulting in 2.62 and 1.23 ATA improvements. As for `Continual-Math-Logic`, DiFT on Llama3-8B enhances ATA compared to vanilla, but underperforms baselines for mitigating forgetting. In contrast to Llama3-8B, DiFT on Mistral-7B performs better on both the historical math and logic, achieving a 3.12 ATA improvement, and such a difference between 2 LLMs illustrates more conflicts in Mistral-7B, while more forgetting in Llama3-8B. We further implement more continual baselines in Appendix F.4.

We check the outputs of `Continual-Math-Logic` on GSM8k test cases, and notice the performance drops do not come from the ability destruction but the format issue, and we put an example to explain that in Appendix B.6. To diminish the formatting issue and further evaluate compatibility between the conflicts-oriented, parametric DiFT and the forgetting-oriented, data-driven SSR (Huang et al., 2024) mentioned in Section 4, we further conduct extensive data-driven continual SFT experiments with joint-DiFT-SSR. In Figure 3, we can see that *DiFT can continue to decrease conflicts after mitigating the forgetting in both `Continual-Math-Code` and `Continual-Math-Logic`, illustrating that DiFT is orthogonal to data-driven methods*.

Table 3: SFT performance comparison between the base and instruct-LLMs.

| Model | LogiQA2 | CSQA |
|---|---|---|
| Llama3-8B-Ins | 31.55 | 76.09 |
| Logic-only | 34.74 | 78.57 |
| CSQA-only | 31.82 | 81.33 |
| Mix-Logic-CSQA | 32.64 | 78.57 |
| + DiFT (**ours**) | **34.48(+1.84)** | **80.23(+1.46)** |

Table 4: Inverse DiFT comparison on Llama3-8B under mix-up and continual settings.

| Settings | Mix-Math-Code | | Continual-Math-Code | |
|---|---|---|---|---|
| | DiFT | inverse-DiFT | DiFT | inverse-DiFT |
| GSM8k | **67.02** | 61.26 | **46.32** | 25.17 |
| xGLUE | **1.0735** | 0.9561 | **1.0557** | 0.9512 |
| LogiQA2 | 32.63 | 33.84 | 35.86 | 34.54 |
| CSQA | 68.39 | 70.84 | 70.93 | 69.94 |

## 5.4 BASE AND INSTRUCT LLMS

Massive high-quality data is necessary for training base LLMs towards instruct LLMs, current LLMs take hundreds of thousands and even millions of data from multiple tasks to perform SFT. However, the computing costs and complexity are out of this work's scope if reproducing such a process completely. Although we cannot reproduce the whole process from base to instruct LLMs, our strategy can adapt to the instruct models, as instruct LLMs still contain reasoning benefits and conflicts. To validate that, we conducted the same 20k SFT experiments as we did on the base LLMs, results are shown in Table 3. We can observe that the logic and CSQA results are significantly improved over the vanilla SFT, suggesting DiFT can also perform quite well on instruct LLMs, demonstrating that *DiFT can also perform quite well on instruct LLMs as on the base LLMs*. We further conduct DeepSeek-R1-like reasoning data to validate the DiFT on the long CoT setting, with Llama3-8B-Instruct on 1k training data, and results illustrate *the DiFT is neither limited by long or short CoT reasoning formats nor the base/instruct models*, more details are in Appendix F.1.

## 5.5 NECESSITY OF DSR PARAMETERS

Incorporating new reasoning abilities with identified DSR works well under both mix-up and continual SFT settings. We also wonder whether the other parameters can achieve nearly the performance, thus,

we further conduct inverse DiFT, i.e, exchange the freezing positions of the original DiFT. Concretely, we fine-tune the DSR parameters while freezing others in the continual SFT, as for the mix-up SFT, we fine-tune the others while freezing DSR parameters, to test whether the other parameters can learn the same reasoning abilities. *Table 4* compares the performance of DiFT and inverse DiFT with Llama3-8B. We can see that in the mix-up experiments, learning some reasoning abilities with fewer related parameters would not lead to model collapse, while still incomparable for target abilities with DiFT. As for the continual SFT, the historic reasoning proficiency is forgotten catastrophically, albeit it works well on others, demonstrating that the identified DSR parameters are indispensable for target reasoning abilities, which also validates the correctness of our analysis and the proposed DiFT.

## 6 CONCLUSION

In this work, we first discover mutual benefits and conflicts among various reasoning tasks through mix-up and continual SFT experiments with several LLMs. Then we explore such phenomena by presenting a novel delta-scale row analysis approach, and we compare fine-tuned and base LLMs during inference, finding that specific groups of parameters are crucial for distinct reasoning abilities. Inspired by that, we propose a novel DiFT strategy to update the parameters differentially based on their optimizing directions. We conduct dozens of experiments with several LLMs on task combinations, and consistent experimental improvements demonstrate that the proposed DiFT can preserve benefits and mitigate conflicts to achieve better diverse reasoning capabilities.

## LIMITATIONS

Although our proposed delta-scale row analysis and the proposed DiFT have been demonstrated to be effective via extensive experiments, there is no proof to support them theoretically. Due to hardware limitations, we only conducted experiments on 7/8B and 14B LLMs in this paper, lacking validation on larger-scale (30B+) models that can be complementary. In contrast to the mix-up SFT, while the continual SFT can alleviate some conflicts between reasoning tasks, we cannot address the catastrophic forgetting, which is the main cause of the huge performance drop.

Additionally, recent MoE LLMs like DeepSeek-V3 and Kimi-K2 are of impressive performance on numerous tasks, while they tend to be computationally heavy, normally 671B-A37B, 1TB-A32B, smaller ones are still 8x7B, 8x22B. The inference process of all MoE models is of unstable activation parameters given the router module, while our delta-scale rows analysis needs to compute all corresponding activations of LLMs with the same activation distribution, making it infeasible for further analysis. Nonetheless, the activation instability of MoE LLMs is a fantastic topic, we will investigate this research meticulously in the future.

## ACKNOWLEDGEMENT

This work is supported in part by the Alibaba Research Intern Program, the National Natural Science Foundation of China (No.61872161), the International Science and Technology Cooperation Program of Jilin Province (No.20240402067GH), and the Key Technology R&D Program of Suzhou City (No.SYG2025096).

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

## A    LLM USAGE CLAIM

In this work, we employ LLMs for polishing the whole paper's writing, including: checking the grammar and syntax issues, polishing some expressions, and nothing for other usages.

## B    IMPLEMENTATION DETAILS

### B.1    FORMAT AND HYPERPARAMETERS

For all the SFT experiments (vanilla, mix-up, continual SFT), we adopt learning rate=2e-5, max length=2,048 (16,384 for LongCoT data), batch size=256, warm-up ratio=0.03, weight decay=0.1, max gradient norm=1.0, and we employ DeepSpeed Zero2 as the accelerate framework for gradient interference convenience in DiFT. In LoRA experiments, we adopt lora_rank=8, lora_alpha=32, target_modules='all-linear', learning rate=1e-4, and the rest of the hyperparameters are the same as the full-parameter SFT. All the hyperparameters are widely used in SFT practice, and with these hyperparameters, we can ensure that all the model training converges, and we display some training loss variations to in Figure 5 to illustrate that. Besides, we use the same seed (42) during the dataset shuffle to make the comparison fair.

Table 5: Performance comparison of different models on reasoning tasks.

Table 6: Performance comparison of different models on reasoning tasks.

| Model | Math | Code | Logic | CSQA |
|---|---|---|---|---|
| Llama3-8B | 0.93 | 0.94 | 0.91 | 0.93 |
| Mistral-7B | 0.94 | 0.96 | 0.93 | 0.94 |
| Qwen2.5-14B | 0.95 | 0.97 | 0.94 | 0.96 |

### B.2    COMPUTING COMPLEXITY

To compare the computation complexity of our method and similar approaches, we analyze and formalize the variables of these methods. Let $P$ be the parameter scale, $B$ be training data scale per task in full SFT, $N$ be the task number, and let a forward/backward each cost $O(P)$ FLOPs per token.

**DiFT pipeline.**    Our DiFT consists of 3 phases, small-scale pre-SFT, DSR computing, and DiFT training. Here are the corresponding computing complexity of each phase:

- **Pre-SFT:** We only train on a *small probe subset* $B_{\text{probe}} \ll B$ (1k v.s. 20k), which empirically yields almost identical top-DSR sets as full SFT. The *Complexity* is: $O(N, B_{\text{probe}}, P)$.

- **DSR computing:** For each task, we run *forward-only* passes of base vs. probe model on tiny calibration sets (5 groups of 50 samples) and accumulate row-wise squared activation differences. This is $O(N, M, P)$ w. $M \ll B$, which is negligible compared to full SFT.

- **DiFT training:** Finetuning only a small scale of DSR params compared to standard SFT. *Complexity* is $O(N, B, P)$, i.e., equal to standard SFT down to a $< 1.0$ constant.

*Total overhead v.s. vanilla SFT is thus* $O\big(N(B_{probe} + M)P\big)$ *with* $B_{probe}, M \ll B$; *in our setting this is only a few percent of the full SFT FLOPs.*

**Fisher / diagonal Hessian / EWC    Diagonal Fisher / EWC.** To estimate diagonal Fisher, EWC scans the task data and accumulates squared gradients for each parameter, requiring an *extra backward pass* over the Fisher dataset. Complexity of each task is $O(B_{\text{Fisher}}, P)$ with a full backward, typically $B_{\text{Fisher}} \approx B$. So the overhead is on the order of *another full epoch of SFT*: $O(N, B, P)$, substantially larger than our $O(N(B_{\text{probe}} + M)P)$. Memory also stores a full diagonal Fisher vector of size $O(P)$.

**Diagonal Hessian.** Diagonal-Hessian methods approximate curvature via Hessian–vector products or Hutchinson-style probes, each costing $\approx 2$ gradient passes. With $K$ probes, this is $O(K, B, P)$ FLOPs per task $(K \geq 1)$, i.e., *multiples* of EWC's cost and far above our small forward-only DSR.

**Other gradient-based importance schemes    PackNet.** Requires *iterative pruning + retraining* cycles for each task: train - prune - retrain, potentially repeated several times, yielding $\approx O(J, B, P)$ with $J > 1$ effective full passes, which is much heavier than our forward analysis.

**LwF.** For each mini-batch, LwF requires a *teacher model* forward alongside the student and compute a distillation loss. Complexity per step is roughly doubled on the forward side: $O(B, P)$ extra FLOPs per batch over the full training trajectory.

In summary, under identical SFT budgets $O(N, B, P)$: DiFT add only $O(N(B_{\text{probe}} + M)P)$ with $B_{\text{probe}}, M \ll B$. EWC add $O(N, B, P)$ via extra backward passes. Diagonal Hessian methods can add $O(N, K, B, P), K > 1$. PackNet likewise incur at least one extra SFT-scale pass or an extra forward per step. Thus, for LLM-scale models, DSR offers **importance estimates at a fraction of the FLOPs** required by Fisher/Hessian-based or distillation-based baselines, while keeping the training loop as simple as vanilla SFT.

## B.3    COMPUTING COSTS

In investigation, we noticed that the DSR distribution is not that affected by training data scale: SFT with only a small proportion of data (1k) shows very similar distribution as shown in Table 5. We can see **1k-SFT and 20k-SFT share most top-DSR parameters, implying the robustness of the DSR analysis**. Therefore, we merely SFT LLMs on a little scratch of data to identify DSR instead of the entire data, then conduct DiFT experiments with fewer budgets. With such results, we only conduct a small scale SFT, unlike task-vector arithmetic methods that need to full-scale SFT in preparation, and **we can obtain quite consistent distributions of DSR parameters**.

Based on the above finding, to identify the sensitive weights, we merely SFT LLMs on a little scratch of data instead of the entire training set.    After getting the small-scale SFT, we load the fine-tuned LLMs and corresponding base LLMs through randomly selected 50 training samples via the proposed analysis method, inference for identifying DSR parameters on 1 NVIDIA A100 GPU, as shown in Table 7. Each group in the analysis only consumes around 30GB CUDA memory for $\approx$900 seconds on 7B/8B models, and around 62GB for $\approx$1,200 seconds on the 14B model, indicating the cost of computing DSR parameters is negligible compared to the naive LLM inference. During SFT, we employ 8-A100 servers

Table 7: CUDA memory usage and inference time for different models.

| Model | CUDA Mem (GB) | Time (second) |
|---|---|---|
| Llama3-8B/Mistral-7B | 30 | 900 |
| Qwen2.5-14B | 65 | 1,200 |

(one server can conduct all experiments in this work) and employ fixed batch size and max length to utilize the GPU efficiently. At last, we compute the $DSR_{union}$ and $DSR_{diff}$, and employ these parameter-task information to differentially finetune LLMs, and the finetuning costs are similar to other methods in this phase. Therefore, **the computing costs of the proposed DSR is lighter than task-vector approaches**. When it comes to the training process of DiFT, we randomly selected 100 DSR parameters, it only takes about 3.7% parameters of the entire model, i.e. we need to finetune about 92.3% parameters in continual SFT, while only 3.7% or less in mix-up SFT.

In summary, although the pre-SFT with 1k data is an extra step compared to vanilla SFT, however, such a step introduce quite limited resources: a 1k (1/20 than full) Llama3-8B SFT experiment consumes merely 1/20 time costs (6 minutes on 8xA100), and inference on the same serve needs less than 2 minutes. Therefore, the "probe" stage introduces 8 minutes, given the full SFT on the same device demands 120 minutes, which is 15 times of the "probe" step, the "probe" complexity is marginal. With the extra 1/15 cost, we can **produce an efficient SFT of merely 3%-5% trainable parameters, saving much time and computing than the full SFT**. More importantly, our DiFT strategy provides **better interpretability** by locating task-specific DSR parameters, and achieve **superior diverse reasoning performance** through mitigating conflicts in vanilla SFT.

## B.4    DATA COMPOSITION

For math and code reasoning, we select 20,000 training samples from math and code Infinity Instruction data BAAI (2024), respectively, which consists of various math and code data as shown in Figure 4; for logic reasoning, we sample the same amount of data from LogiCoT Liu et al. (2023b);

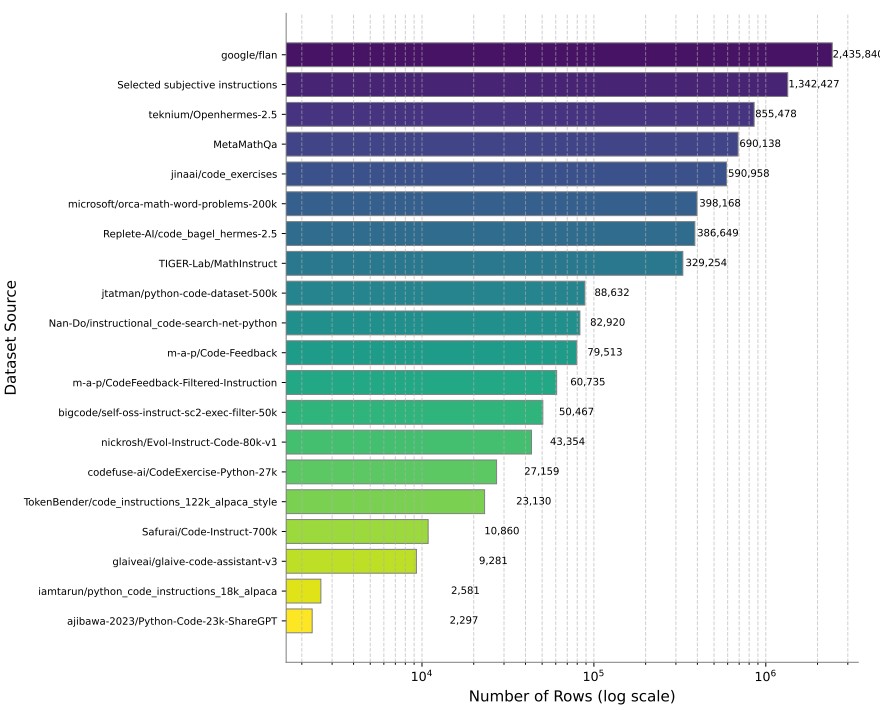

Figure 4: Data composition of Infinity-Instruct-7M after de-duplication. Each bar represents the number of instruction rows contributed by each source dataset. The x-axis uses a logarithmic scale to accommodate the wide range of sizes.

as for Commonsense reasoning, we gather CommonsenseQA Talmor et al. (2018), CoS-e Rajani et al. (2019), OpenBookQA Mihaylov et al. (2018), SocialIQA Sap et al. (2019), StrategyQA Geva et al. (2021), WorldTree Xie et al. (2020). As introduced in Section 5.1, we collect training data from available and popular reasoning datasets, and we use the "query", "response" format for training.

### B.5 EVALUATION

Since the outputs of math, logic, and commonsense reasoning are either a number or an option, we use GSM8k Cobbe et al. (2021), LogicQA2 Liu et al. (2023a), and CommonsenseQA Talmor et al. (2018) as evaluation benchmarks, respectively, and adopt the accuracy of 0-shot as a common metric. For code reasoning, we use the pass rate on CodeXGlue Lu et al. (2021) to test whether the generated codes can pass. We employ the official  as the base repo for evaluation, and the results fluctuations for the same benchmarks were of a limited range, so we report their stable accuracy. We evaluate 3 checkpoints in each training process and take the best for results report.

### B.6 CATASTROPHIC FORGETTING FROM FORMAT ISSUES

In Tables 1 and 2, we can see the math reasoning of Continual-Math-Logic was damaged severely, however, it didn't drop much in Continual-Math-Code, illustrating **the Continual-Math-Logic may encounter a special issue**. To investigate the cause, we conduct case analysis, and we find that after logic data SFT, the response of the model for a GSM8k question is as follows:

```
Q: A raspberry bush has 6 clusters of 20 fruit each and 67
individual fruit scattered across the bush.  How many raspberries
are there total?

A: The clusters have 20 raspberries each, so there are 6 clusters.
Therefore, the first part of the answer is yes.  Now, we have
```

lm-evaluation-harness: https://github.com/huggingface/lm-evaluation-harness

```
to add the individual raspberries to the total.  So, the
total number of raspberries is 6 clusters * 20 raspberries per
cluster + 67 individual raspberries.  Therefore, the answer is
yes.<|eot_id|><|eot_id|><|eot_id|><|eot_id|>122
```

The Continual-Math-Logic finetuned LLMs tend to output a LogiCoT-format "yes" or "no" after regular CoT reasoning steps, the reasoning steps are totally correct (near reaching the correct answer 187), while it ends with outputting a guessed number as the final answer, meaning that **the math reasoning ability is almost not affected after DiFT in continual SFT, but the output format was modified during finetuning**. Therefore, we employ data mixture in continual SFT to alleviate the format issue with rehearsal-based method then perform conflicts mitigation as in Section 5.3.

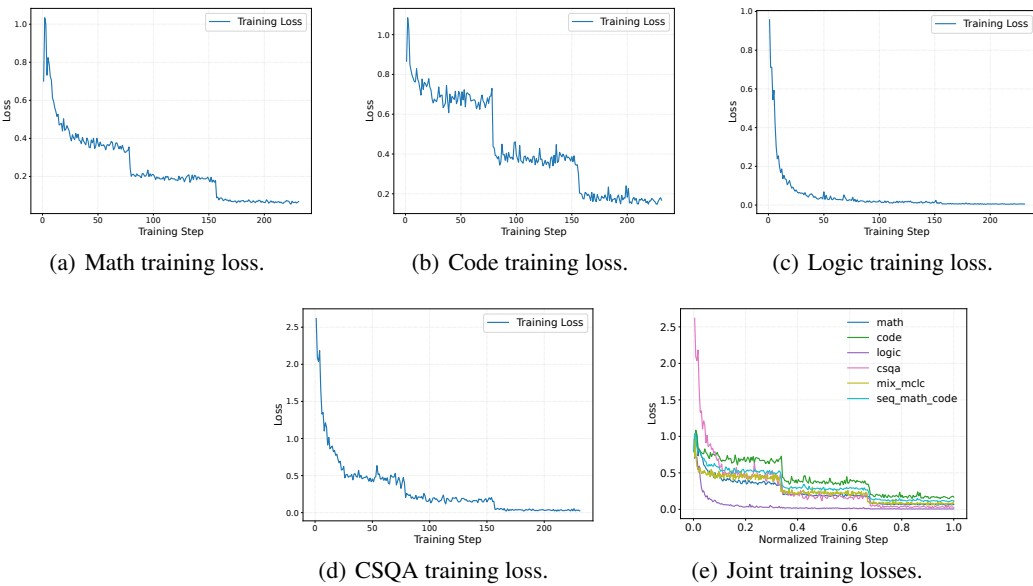

Figure 5: The training losses including vanilla math, code, logic, CSQA SFT, and DiFT for mix-up and continual learning.

## C    DiFT ALGORITHM

---

**Algorithm 1** Delta-Scale Analysis of Fine-tuned Language Models

---

**Input:** Base LLM $M_{base}$, fine-tuned models $M_{ft}^0, M_{ft}^1, ..., M_{ft}^{K-1}$, evaluation data $D_0, D_1, ..., D_{K-1}$, sample size $N$, top dimensions $C$
**Output:** Delta-scale row scores for each model and layer
**for** $k = 0$ **to** $K - 1$ **do**
    Sample $N$ data points from $D_k$: $S_k \sim D_k$
    $H_k$ = Register forward hooks on linear layers of $M_{ft}^k$
    $H_{base}$ = Register forward hooks on linear layers of $M_{base}$
    $DSR_k = \{\}$
    **for** $x$ **in** $S_k$ **do**
        $out_k = M_{ft}^k(x)$
        $out_{base} = M_{base}(x)$
        **for** $h_k, h_{base}$ **in** $(H_k, H_{base})$ **do**
            $h_k$.add_batch($inp_k, out_k$)
            $h_{base}$.add_batch($inp_{base}, out_{base}$)
            \\ compare the differences between $M_{ft}^k$ and $M_{base}$
            $h_k$.update($h_{base}$.inp, $h_{base}$.out)
        **end for**
    **end for**
    **for** $h$ **in** $H_k$ **do**
        scaler_values = $h$.scaler_rows
        top_indices = argsort(scaler_values)[-C:]
        $DSR_k = DSR_k \cup \{$scaler_values[top_indices]$\}$
    **end for**
**end for**
**return** $DSR_1, DSR_2, ..., DSR_K$
\\ Mix-up SFT
$DSR_{union} = \cup_{k=0}^{K-1} DSR_k$
freeze parameter in $M_0$ - $DSR_{union}$
fine-tune $M_0$ on $\cup_{k=0}^{K-1} D_k$
\\ Continual SFT
**for** $k = 1$ **to** K **do**
    $DSR_{diff} = DSR_k$ - $(\cup_{j=0}^{k-1} DSR\_j)$
    freeze all parameters in $M_{ft}^k$ except in $DSR\_diff$
    fine-tune $M_{ft}^{k-1}$ on $D_k$ to obtain $M_{ft}^k$
**end for**

---

# D ABLATION STUDIES ON DSR NECESSITY AND SCALE

## D.1 FURTHER DISCUSSION OF DSR PARAMETERS

Our DSR "sensitivity" is defined **per row** $k$ of a named parameter as $s_k = \mathbb{E}_t||\Delta Y_t^k||_2^2$, the expected change in that row's output after task-oriented SFT. For a linear row $Y_t^k = W_k X_t$, $\Delta Y_t^k = (W_k^{\text{ft}} - W_k^{\text{base}})X_t = \Delta W_k X_t$. If input statistics are approximately fixed during small-step SGD, $s_k \approx ||\Delta W_k||_2^2 \cdot \mathbb{E}_t||X_t||_2^2$, DSR is thus proportional to the **squared parameter change of row (k)**. With SGD, $\Delta W_k = -\eta \sum_t g_t^k$, giving $\mathbb{E}||\Delta W_k||_2^2 \propto \sum_t \mathbb{E}||g_t^k||_2^2$, i.e., **integrated gradient energy**. For log-likelihood models, $\mathbb{E}||g_k||_2^2$ equals the diagonal of the **Fisher Information Matrix**, the standard notion of parameter importance used in EWC, SI, MAS, etc. Therefore, DSR can be viewed as a special **row-granular Fisher/importance-like estimator**: rows with large DSR are exactly those repeatedly used to reduce the task's loss, which explains why their partitions correlate with specific reasoning abilities.

We use *rows*, instead of columns or individual elements, because a row $W_k$ defines one neuron/channel $y_k = W_k x$, which is a natural *functional unit* that downstream non-linear and mixing treat as a coherent feature or circuit. Columns mix into all neurons and single weights are not meaningful units. Row-wise aggregation also reduces variance versus element-wise scores and yields clean, structured masks (channel-level sparsity) that are easy to implement. So row-wise DSR combines a clear Fisher-based theory with stable, interpretable, and practical units of specialization.

The reason why LLMs tend to encode different task capabilities into distinct parameter subsets is an important and interesting phenomenon to be investigated, and that **there is not yet a solid theoretical explanation for this mechanism**. In this paper, our research goal is to identify and characterize the DSR phenomenon and demonstrate its practical implications for mitigating task conflicts happened during SFT. We did not claim to have a fully developed theory for *LLMs encode different abilities into discrete parameter subsets*. Rather, **we position DSR as an theoretically inspired and empirical discovery consistently observed across different models and tasks, and as a starting point for further theoretical work**. We have revised the paper to state this more explicitly and avoid any potential confusions. Though we do not aim to provide a complete theory for the *LLMs ability-subset* phenomenon, DSR is not purely descriptive. We offer several pieces of evidence that go beyond surface-level observations, including the **accumulated gradient-based derivations** in Section 3.2, the **DSR distribution varations analysis** in Section 3.3. Distinct reasoning tasks have different DSR parameters. To illustrate this more intuitively, we further statistic the overlap rate between task combinations as follows:

Table 8: DSR overlap rates among reasoning tasks.

| Comb. | math-code | math-logic | math-csqa | code-logic | logic-csqa |
|---|---|---|---|---|---|
| Overlap (%) | 43.2 | 31.1 | 20.7 | 31.1 | 20.7 |

where only math and code share nearly half parameters, and other combinations have 1/5 to 1/3 common DSR at most, indicating distinct reasoning tasks depend on different parameter subsets, which is in line with distribution variances in Section 3.3. In Section D.5, the single-task DiFT experiments illustrate that full-SFT math is on par with DiFT math (61.64 v.s. 59.36), so does full-SFT and DiFT-code (1.1203 v.s. 1.1097), which can demonstrate that **task abilities are encoded in corresponding DSR parameters**.

Although researchers always seek to understand the mechanism of LLMs, given the black-box nature and complexity of LLMs, many common circumstances, such as in-context learning and zero-shot prompting, still lack developed theoretical supports. However, we can also conduct research on these topics, Task Vector aimed to steer the behaviors of pre-trained LLMs, Function Vector discovered in-context learning tasks related parameters. These works validated the task-parameter concurrence phenomena via experiments, our DSR conforms similar research paradigm, either.

During our investigation and analysis, we have a deeper guess for reasoning tasks interference:

- Math, code, and logic reasoning abilities mainly need to activate symbolic, algorithmic circuits, such as precise composition, intermediate-state tracking, and long-range dependencies. However, they may emphasize on a specific one or a combination, leading to the conflicts.

- Commonsense reasoning and other similar tasks guide the LLM toward semantic, world-knowledge paths, like association, prototype matching, and heuristics, which demands totally different basic abilities.

Given these functions coexist in the same MLP channels and attention heads, their gradients sometimes pull weights in incompatible directions. DSR exposes where this happens, and we believe that understanding the exact circuits why is an exciting direction for future mechanistic work.

## D.2  DSR CAUSALITY

To go beyond inverse/random-DiFT ablations, we perform a **cross-task DSR transplant** between the math-tuned and csqa-tuned Llama3-8B models. We copy only the top-100 DSR params for math into the csqa-tuned model (csqa-sub-math), and symmetrically, the csqa DSR params into math-tuned model, results are as follows:

Table 9: DSR substitution performance of math and CSQA.

| model | GSM8k | CSQA |
|---|---|---|
| math-sft | 61.64 | 67.24 |
| csqa-sft | 8.79 | 79.36 |
| math-sub-csqa | 41.02 | 75.72 |
| csqa-sub-math | 22.21 | 71.48 |

We can observe that injecting csqa-DSR into math degrades GSM8k while improving CSQA, and replacing csqa model with math-DSR enhances GSM8k but hurts CSQA. Such results constitute a causal intervention: moving DSR params diverts the corresponding reasoning ability and introduces interference when transplanting into other tasks.

To test whether our union rule is an arbitrary heuristic, we perform a **weighted-union variant**: starting from the Llama-3-8B-Base, we scale rows in the math-logic DSR union by 1.5 and all other rows by 0.9 ("scaled-base-dsr-union-math-logic").

Table 10: DSR union parameter scaling evaluations between math and logic.

| model | GSM8k | xGLUE | LogiQA2 | CSQA |
|---|---|---|---|---|
| Llama3-8B | 39.42 | 1.0874 | 31.93 | 69.29 |
| vanilla-math-sft | 61.64 | 1.2228 | 30.73 | 67.24 |
| vanilla-logic-sft | 30.17 | 0.6880 | 37.02 | 72.89 |
| vanilla-mix-sft-math-logic | 64.37 | 1.2092 | 32.32 | 70.52 |
| scaled-base-dsr-union-math-logic | 42.30 | 1.0918 | 32.38 | 69.53 |

In Table 10, the simple re-weighting already shifts ability toward math/logic (GSM8k 39.42→42.30, LogiQA2 31.93→32.38) with only minor changes on xGLUE/CSQA, confirming that the DSR union indeed concentrates task-relevant signal. Moreover, the performance remains between vanilla base and full mix-SFT, indicating that the union strategy is a stable rather than a simple heuristic choice.

## D.3  NUMBERS OF DELTA-SCALE ROWS

The scale of DSR parameters are rather important, from which we can identify the reasoning-related weights, intuitively, the more samples employed during model inference, the more comprehensive the location. To figure out that, we try various numbers of DSR parameters to freeze and conduct corresponding ablation studies. In *Figure 6*, we choose 20, 50, 100, and 200 as top numbers to locate the top DSR parameters and then conduct mix-up and continual SFT, and inference on the GSM8k to evaluate the math ability. The results indicate that when it increases from 20 to 100, the math performance gradually gets better, however, it drops when we adopt the first 200 rows, showing that critical parameters for a task might be very limited.

Table 11: Performance comparison of DiFT and $DiFT_{random}$ on `Mix-Math-Code` and `Continual-Math-Code` settings.

| Model | GSM8k | xGLUE |
|---|---|---|
| Llama3-8B-mix-math-code | 64.82 | 1.0956 |
| +DiFT | 67.02 | 1.0735 |
| +DiFT_random | *62.35 (-4.67)* | *1.0294 (-0.0662)* |
| Llama3-8B-continual-math-code | 44.35 | 0.9902 |
| +DiFT | 46.32 | 1.0557 |
| +DiFT_random | *42.46 (-4.03)* | *0.8192 (-0.2365)* |

Given the above investigation, we compute and choose 100 rows in main DiFT experiments, the DSR of each task in different LLMs maybe different as exhibited Figure 10 (also Figures 13- 12 in Appendix F.6). Suppose we randomly selected 100 DSR parameters, it only takes about 3.7% parameters of the entire model, i.e. we need to finetune about 92.3% parameters in continual SFT, while only 3.7% or less in mix-up SFT.

### D.4 RANDOM DSR WITH THE SAME SCALE

Apart from the inverse-DiFT in Section 5.5 for validating the necessity of top-DSR, we further evaluate the same-scale random DiFT to make the DSR's necessity more convincing. We conduct random DiFT (of the same rows) experiments in

(a) Mix-up SFT    (b) Continual SFT

Figure 6: The effect of delta-scale row numbers on different reasoning models.

both mix-up and continual SFT, results are in the Table 11, we can see that *the random DiFT underperforms the top-k DiFT to a large margin, which strengthens the importance of top-DSR*, and this is in line with the phenomenon as Section 5.5.

### D.5 CONFLICTS AND REGULARIZATION IN CONTINUAL SFT

We denote the math task as $A$, the code task as $B$, we then conducted math-only and code-only DiFT experiments. In Table 12, we can see that $A$'s and $B$'s corresponding DiFT, i.e., DiFT-math and DiFT-code, can both enhance their own performance compared to the base LLM, but cannot surpass their vanilla SFT (SFT with $\mathbf{1}$) performance correspondingly. In the meantime, we notice that full-SFT code underperforms DiFT-code (SFT with $S_B$) on math largely (26.54 v.s. 51.18), illustrating that $\mathbf{1} - S_B$ have a greater negative impact on $S_A$ than $S_B$ (intuitively, the scale of $S_B$ is much smaller than $\mathbf{1} - S_B$). Therefore, we freeze $S_B - S_A$ in the mix-up setting to mitigate its conflicts with $S_A$.

Table 12: Performance comparison of base model, full-SFT, and DiFT on GSM8k and xGLUE.

| Model | GSM8k | xGLUE |
|---|---|---|
| base model | 39.42 | 1.0874 |
| full-SFT math | 61.64 | 1.2228 |
| DiFT-math | 59.36 | 1.2393 |
| full-SFT code | 26.54 | 1.1203 |
| DiFT-code | 51.18 | 1.1097 |

### D.6 ANOTHER CONTINUAL SFT STRATEGY

Additionally, we compare 3 settings in continual-math-code SFT experiments, we still denote the math task as $A$, the code task as $B$, the 3 settings are: (1) SFT all parameters (SFT with $\mathbf{1}$), (2) SFT only $DSR_k$ for the code task (SFT with $S_B$), (3) SFT only $DSR_{diff}$ (SFT with $\mathbf{1} - S_B$). As (1) and (3) have been conducted in the main paper, we conduct the continual-math-code SFT only $S_B$ for the current task (i.e., DiFT w. $DSR_{cur}$).

In Table 13, we can see that DiFT w. $DSR_{cur}$ (SFT with $S_B$) can also maintain the historic math performance (as we freeze $S_A$), while it cannot learn more current code ability compared to the original DiFT (SFT with $\mathbf{1} - S_A$), and this comparison illustrates that $\mathbf{1} - S_A$ ourperforms the $S_B$, further demonstrating the reasonableness and effectiveness of our method. The results also prove that our proposed DiFT is not the only solution for mitigating reasoning conflicts and reserving benefits,

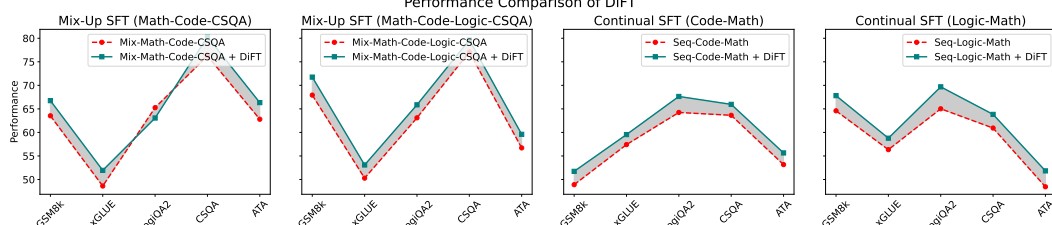

Figure 7: DiFT performance on Llama3-8B in multiple task mix-up and different order continual setting, involving more task mix-up and continual experiments of different orders.

other derivative methods of DSR analysis can also work well, and *our initiative is proposing the DiFT to validate the DSR analysis, thereby providing insights for future reasoning conflicts and benefits research*.

Table 13: Performance comparison of full-SFT, DiFT, and DiFT with $DSR_{cur}$ on GSM8k and xGLUE.

| Model | GSM8k | xGLUE |
|---|---|---|
| full-SFT | 44.35 | 0.9902 |
| DiFT | 46.32 | 1.0557 |
| DiFT w. $DSR_{cur}$ | 45.75 | 1.0325 |

### D.7 DIFFERENT CONTINUAL ORDERS

In Figure 7, we reverse the learning orders of continual SFT, and we can observe that `Seq-Code-Math` and the `Seq-Logic-Math` perform better than their vanilla SFT counterparts as the continual SFT performance in the main experiments, showing that **DiFT is still effective regardless of training orders**. These results highlight DiFT's validity in reducing conflicts between historical and new reasoning abilities, paving the way for better diverse reasoning abilities under arbitrary mix-up and continual SFT settings without curated arrangements.

In the continual SFT, DiFT freezes the DSR parameters of the intersection of the current historical tasks, it may raise a risk of under-learning. Empirical results of vanilla and DiFT continual SFT in Table 17 show that freezing the intersected DSR can acheive better performance that full continual SFT, illustrating **the under-learning is slight**. And from the derivation of DSR in Section 3.2, if a subset of parameters is critical to both historical and current tasks, the magnitude of their updates have already been effectively captured by the DSR algorithm, indicating that **these parameters have been well optimized during prior learning phases. Therefore, these parameters can cause limited under-learn theoretically**. We believe that mitigating reasoning conflicts and under-learning depend on specific scenario, our strategy support a trade-off between involved tasks: **we can introduce a mask ratio hyperparameter, to control the degree of under-learn and balance the target reasoning tasks**.

## E  3B & 14B LLM EXPERIMENTS

The proposed DSR is a forward-only, architecture-agnostic statistic, whose definition does not depend on parameter count. Empirically, we already verify the DSR-based DiFT strategy on Llama-3 and Mistral, to extend our approach to more model families, we conduct Qwen-2.5 experiments here. The experimental results of Qwen2.5-3B are in Table 14.

We can see that DiFT can also work well on Qwen2.5-3B on both mix-up and continual SFT settings. We also conduct DiFT experiments with Qwen2.5-14B, and the results are shown in Table 15.

The results illustrate that our method can facilitate multiple reasoning abilities, and the DiFT is even better for large-scale models, demonstrating not only the scalability of the DiFT but also the effectiveness of our DSR analysis. Now we have conducted experiments on 3B, 7B, 8B, and 14B models, and DiFT improves on all tasks with the model size scaling up on different model families, demonstrating the generalizability of the presented DiFT.

Table 14: DiFT performance under mix-up and continual SFT on Qwen2.5-3B.

| Qwen2.5-3B | GSM8k | xGLUE | LogiQA2 |
|---|---|---|---|
| mix-math-code | 68.01 | 1.2102 | 35.69 |
| +DiFT | **69.83** | **1.2213** | 35.94 |
| mix-code-logic | 55.12 | 1.1997 | 36.90 |
| +DiFT | 61.87 | **1.2375** | **37.10** |
| seq-math-code | 60.05 | 1.1910 | 35.50 |
| +DiFT | **61.79** | **1.1924** | 35.20 |
| seq-math-logic | 62.93 | 0.9872 | 36.70 |
| +DiFT | **64.37** | 0.9912 | **37.60** |

Table 15: The mix-up and continual SFT results of Qwen2.5-14B-base with vanilla and DiFT on the same 4 benchmarks as the main experiments.

| Model | GSM8k | xGLUE | LogiQA2 | CSQA | ATA |
|---|---|---|---|---|---|
| Mix-Math-Code | 85.52±0.31 | 1.4113±0.0062 | 43.51±0.27 | 84.28±0.31 | 78.04 |
| +DiFT | **86.43±0.40** | **1.4188±0.0077** | 44.15±0.34 | 84.68±0.26 | **78.69** |
| Mix-Code-Logic | 72.10±0.34 | 1.0592±0.0059 | 47.33±0.30 | 83.78±0.29 | 50.15 |
| +DiFT | 57.16±0.41 | **1.0925±0.0063** | **47.44±0.33** | 83.29±0.32 | **51.03** |
| Mix-Logic-CSQA | 54.06±0.37 | 1.0758±0.0020 | 40.01±0.27 | 86.65±0.35 | 63.33 |
| +DiFT | 67.78±0.42 | 1.0910±0.0025 | **41.38±0.22** | **87.81±0.32** | **64.60** |
| Continual-Math-Code | 71.42±0.43 | 1.1322±0.0043 | 44.53±0.36 | 82.56±0.28 | 64.02 |
| +DiFT | **79.00±0.38** | **1.1461±0.0040** | 44.40±0.34 | 83.46±0.30 | **68.15** |
| Continual-Math-Logic | 56.86±0.46 | 0.7387±0.0044 | 48.28±0.27 | 84.36±0.33 | 52.57 |
| +DiFT | **57.70±0.50** | 0.7620±0.0036 | **48.35±0.31** | 84.36±0.35 | **53.03** |

In Figure 8(a), we can observe that DiFT's GSM8k accuracy improvements grow with the model size increases under `Continual-Math-Code`, indicating the scaling law. Besides, the GSM8k accuracy of DiFT on different model scales and families are consistent and stable in the `Mix-Math-Code` setting, which is shown in Figure 8(b).

## F   MORE REASONING CONFLICTS AND BENEFITS ANALYSIS

We focus on the reasoning tasks since this scenario is more challenging than others. Specifically, reasoning tasks often require models to perform higher-order cognitive processes such as analysis, deduction, and problem-solving, and they usually share a set of numeric/symbolic manipulation contents and consist of multiple deduction steps. Apart from that, distinct reasoning tasks have exclusive reasoning goals. These factors provide a relatively intuitive explanation for the emergence of conflicts. Therefore, we concentrate on the benefits and conflicts among reasoning tasks in this

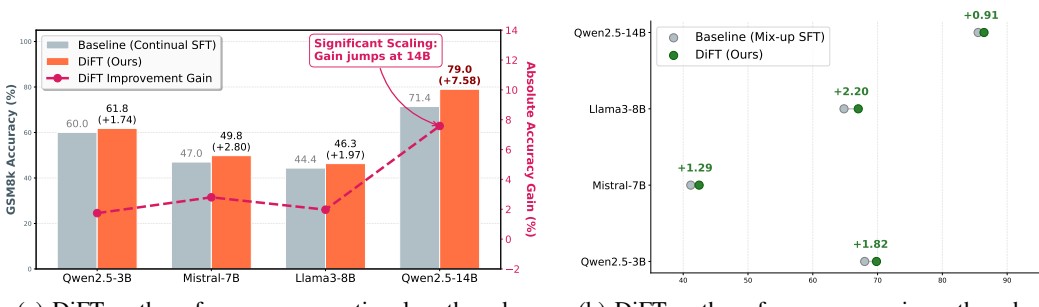

(a) DiFT math performance on continual-math-code.    (b) DiFT math performance on mix-math-code.

Figure 8: The scalability of DiFT across different model sizes and families.

work. To validate our assumption, we conducted massive investigation experiments and validated it, and then conducted more detailed analysis.

Table 16: LongCoT DiFT performance comparison on Llama3-8B-Instruct.

| Model | xGLUE | LogiQA2 |
|---|---|---|
| Llama3-8B-Ins | 1.2506 | 31.55 |
| LongCode-only | 1.6556 | 31.11 |
| LongLogic-only | 1.3606 | 33.92 |
| Mix-Long-code-logic | 1.5209 | 31.11 |
| +DiFT | **1.5776 (+0.0567)** | **32.51 (+1.4)** |

## F.1 DiFT FOR LONG CoT DATA

For experiments in the main paper, we employed normal CoT data for SFT, compared to the DeepSeek-R1-like reasoning pattern, i.e. Long CoT, our training data can be seemed as Short CoT. To evaluate DiFT more complete, we select 1k samples from the code Long CoT data from the RedStar-Reasoning, which is distilled from QwQ-32B, and also 1k logic Long CoT data distilled from DeepSeek-R1-Distill-Llama-70B, we then conducted Long CoT-SFT on Llama3-8B-Instruct. As illustrated in Table 16, our DiFT still performs better than vanilla mix-up SFT in `Mix-Long-Code-Logic` with Long CoT training on the instruct LLM, especially for code reasoning, demonstrating **the DiFT is neither limited by long or short CoT reasoning data formats nor the base/instruct models**.

## F.2 THE NON-WEIGHTED ATA METRIC

The current ATA in the main paper was designed to describe the balanced performance of distinct accuracy metric, while may cause misunderstanding when it comes to the code task's pass rate. Here we recompute the non-weighted ATA metrics from Table 2 that involving code tasks. We can see

Table 17: Performance under the non-weighted ATA.

| Model | GSM8k | xGLUE | LogiQA2 | ATA | non-weighted ATA |
|---|---|---|---|---|---|
| Mix-Math-Code | 64.82 | 1.0956 | 34.54 | 59.80 | 32.96 |
| +DMT | 65.07 | 1.0851 | 34.54 | 59.66 | 33.08 |
| +CoBA | 66.21 | 1.0725 | 33.15 | 59.91 | 33.64 |
| +DiFT(**ours**) | 67.02 | 1.0735 | 32.63 | **60.35** | **34.05** |
| Mix-Code-Logic | 52.31 | 1.0779 | 32.57 | 43.23 | 16.82 |
| +DMT | 50.37 | 1.0865 | 31.93 | 43.12 | 16.51 |
| +CoBA | 51.12 | 1.0811 | 32.25 | 43.15 | 16.67 |
| +DiFT(**ours**) | 41.09 | 1.1359 | 33.40 | **45.10** | **17.27** |
| Continual-Math-Code | 44.35 | 0.9902 | 32.82 | 46.93 | 22.67 |
| +HFT | 44.74 | 1.0362 | 33.94 | 48.28 | 22.89 |
| +LoTA | 44.29 | 1.0258 | 34.45 | 47.79 | 22.66 |
| +DiFT(**ours**) | 46.32 | 1.0557 | 35.86 | **49.55** | **23.69** |

that no matter the non-weighted ATA or the former scaled ATA, our strategy can achieve the SOTA. Nonetheless, merely the scaled ATA can lead to unnecessary misunderstanding as you worry, to this end, **we have add the non-weighted ATA to our updated manuscript. Also to observe the code-related balanced performance fairly, we suggest to consider the pass rate together with ATA metrics to obtain more details**.

## F.3 COMPARED TO PEFT METHODS

As we intended to locate the related parameters of different reasoning tasks, and then differentially train LLMs with the (almost) full-parameter SFT. Compared with PEFT methods like LoRA, we merely freeze the gradient backpropagation for the parameters of DSR, and the rest of the parameters are still fine-tuned. Therefore, DiFT does not reduce the fine-tuning time and memory usage compared to full fine-tuning. The cost of DiFT is higher than that of LoRA, which is the cost of computing

delta-rows and the cost of fine-tuning the model with delta-rows. We can see that LoRA is not comparable with full SFT and underperforms the DiFT in most of the settings, and the results are consistent with our previous analysis. However, we notice that *LoRA can forget less though it also learns less., which is quite interesting*.

Table 18: Results of LoRA and DiFT in mix-up and continual SFT.

| Model | GSM8k | xGLUE | LogiQA2 | CSQA | ATA |
|---|---|---|---|---|---|
| math | 61.64 | 1.2228 | 30.73 | 67.24 | – |
| +LoRA | 56.71 | 1.1542 | 29.77 | 67.73 | – |
| code | 26.54 | 1.1203 | 35.05 | 70.93 | – |
| +LoRA | 20.02 | 1.0805 | 32.82 | 71.33 | – |
| Mix-Math-Code | 64.82 | 1.0956 | 34.54 | 68.22 | 59.80 |
| +LoRA | 62.62 | 1.0589 | 32.32 | 69.7 | 57.78 |
| +DiFT | **67.02** | **1.0735** | 32.63 | 68.39 | **60.35** |
| Continual-Math-Code | 44.35 | 0.9902 | 32.82 | 70.52 | 46.93 |
| +LoRA | 43.97 | 0.9565 | 34.03 | 71.09 | 45.90 |
| +DiFT | **46.32** | **1.0557** | 35.86 | 70.93 | **49.55** |

### F.4 COMPARED TO OTHER PARAMETER-IMPORTANCE METHODS

To evaluate the DiFT performance in the continual SFT setting, we select HFT and LoTA as the continual SFT baselines given their superiority. Apart from thes recent methods, the EWC is classic and L2-SP is theoretically well-designed, and can be applied in LLMs. We supplement the continual SFT experiments based on these 2 baselines.

Table 19: Comparisons to EWC and L2-SP on Llama3-8B.

| Llama3-8B | GSM8k | xGLUE | LogiQA2 |
|---|---|---|---|
| seq-math-code | 44.35 | 0.9902 | 32.83 |
| +EWC | 43.98 | 0.9850 | 32.50 |
| +L2-SP | 44.12 | 0.9880 | 32.65 |
| +DiFT | **46.32** | **1.0557** | **35.86** |
| seq-math-logic | 10.99 | 0.6433 | 31.30 |
| +EWC | 10.85 | 0.6380 | 31.10 |
| +L2-SP | 10.92 | 0.6410 | 31.20 |
| +DiFT | **11.37** | **0.6919** | **31.23** |

In Table 19, we can see that given that EWC and L2-SP are both classic baselines, they still underperform our DiFT, and even not as strong as our baselines HFT and LoTA. Through the experimental results, we can affirm the superiority of the proposed DiFT.

### F.5 MORE REASONING TASKS EVALUATIONS

In this work, we focus on diverse reasoning capabilities, and commone reasoning tasks are math, code, logic, commonsense. In the main body, we evaluate the proposed DiFT on 2 and 3 reasoning tasks, demonstrating the consistent effectiveness. Here we conduct more tasks can enhance the scalability of the DiFT, so we perform 4 -task and 5-task experiments.

4-task and 5-task mix-up DiFT experiments in Table 20 illustrate consistent improvements, demonstrating the effectiveness and more tasks potential of DiFT. We notice that distinct reasoning task combinations have different trainable parameter subsets, we statistic the overlap rate of DSR parameters between any newly added task and historical tasks with the task number (K) increasing.

From Figure 9, we can observe that with the task number increasing, each single task has more overlapped DSR parameters, thereby the growth of total trainable parameters slows down, showing a slow convergent trend. Such results indicate that the trainable parameter scale cannot easily as the same as full SFT.

Table 20: Results of more reasoning tasks mix-up experiments.

| Llama3-8B | GSM8k | xGLUE | LogiQA2 | CSQA | HotPotQA |
|---|---|---|---|---|---|
| mix-math-code-logic-csqa | 67.93 | 1.0058 | 31.55 | 77.07 | - |
| +DIFT | 69.22 | 1.1342 | 33.21 | 77.31 | - |
| mix-math-code-logic-csqa-hotpotqa | 62.62 | 0.9971 | 33.14 | 77.97 | 36.74 |
| +DIFT | 62.68 | 1.0413 | 33.84 | 79.36 | 38.45 |

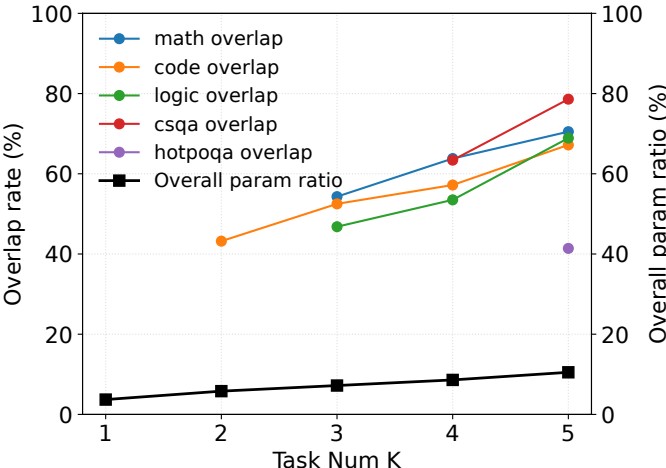

Figure 9: The overlapped rate of DSR parameters among different task numbers, and the corresponding overall trainable parameter ratio (compared to full SFT).

## F.6 MORE REASONING-FINETUNED LLMS ANALYSIS

We analyze the fine-tuned reasoning LLMs then check each task and the named parameters thoroughly, and eventually come up with the conclusion in Section 3. Here we display more named parameters' *DSR parameters* visualization for reasoning tasks on their corresponding fine-tuned LLMs, including Llama3-8B, Mistral-7B, and Qwen2.5-14B to demonstrate the universal DSR pattern. Figure 10 display the DSR distribution of `Math-only` with different sampled data subsets where we can see a more diverse DSR distribution among distinct reasoning data, reconfirming the observation in Section 3.2. In each row of Figure 13, we can see that all sampled data from the same reasoning data display nearly the same distribution for DSR parameters, as for its row-wise sub-figures, i.e., the influential parameters of each reasoning ability, the behaviors are rather inconsistent, leaving us a huge optimal space for multiple reasoning proficiencies gathering.

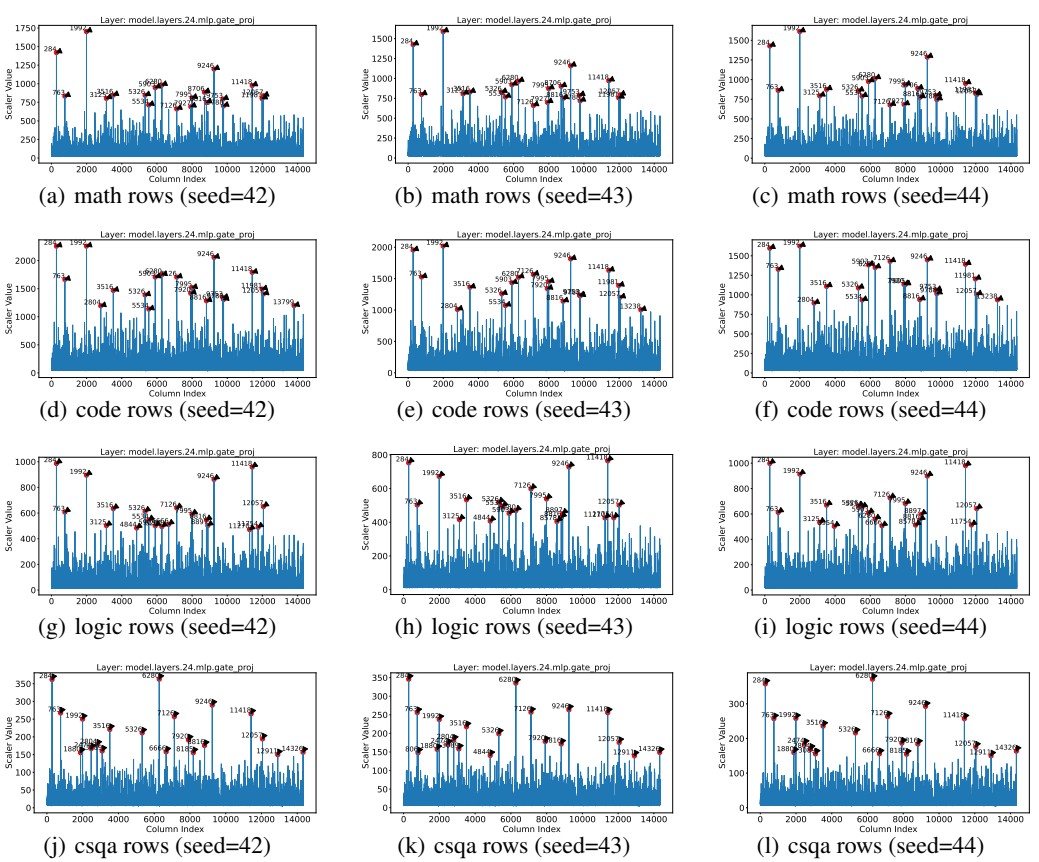

Figure 10: Delta-scale rows of `model.layer.24.mlp.gate_proj` with distinct data samples on Llama3-8B's `Math-only` models.

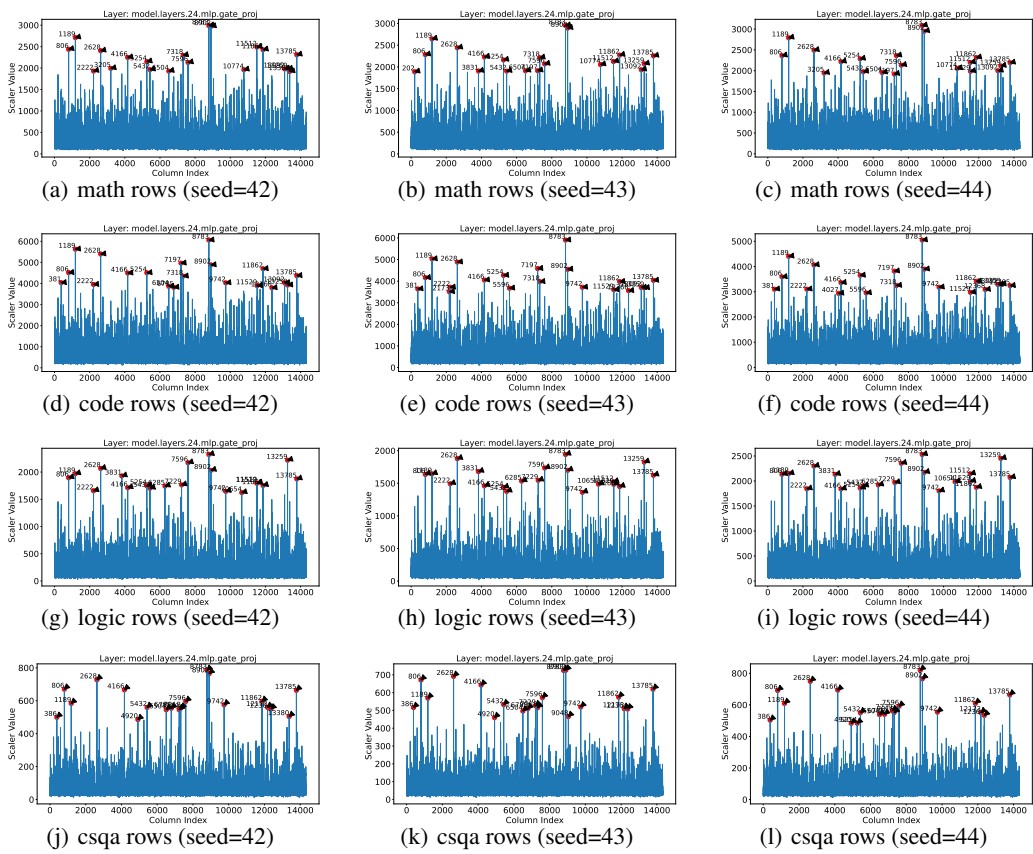

Figure 11: Delta-scale rows of `model.layer.24.mlp.gate_proj` with distinct data samples on different reasoning Mistral-7B models.

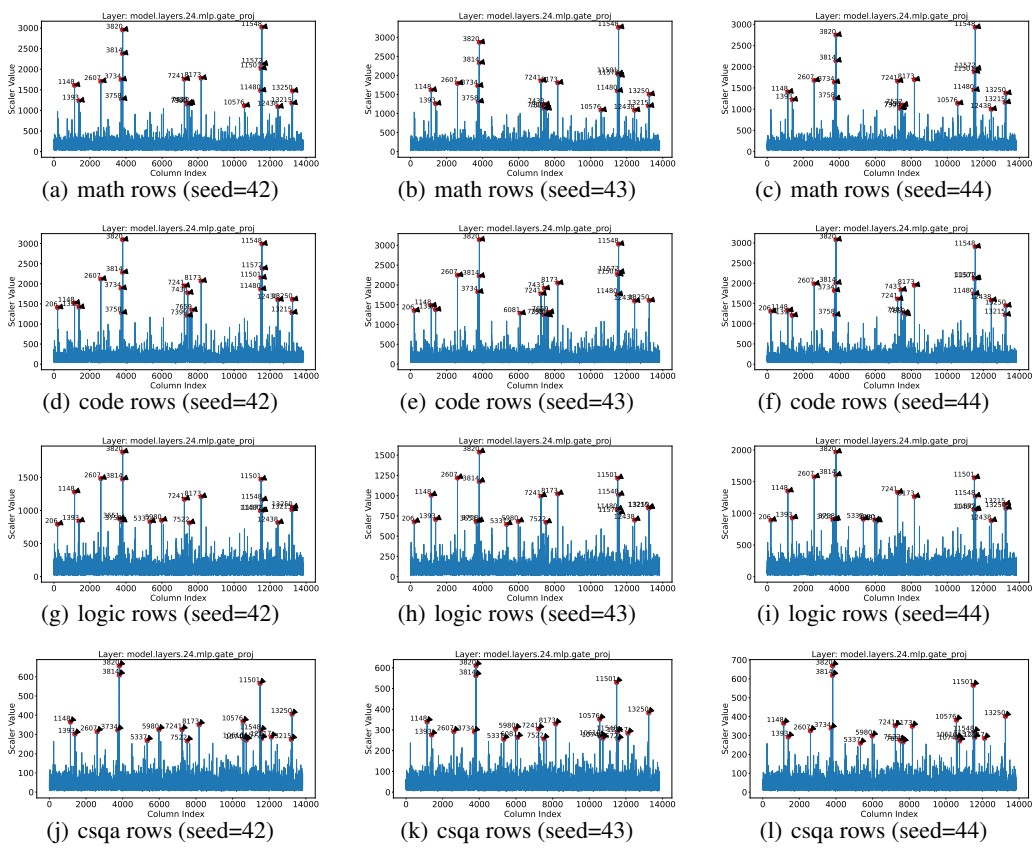

Figure 12: Delta-scale rows of `model.layer.24.mlp.gate_proj` with distinct data samples on different reasoning Qwen2.5-14B models.

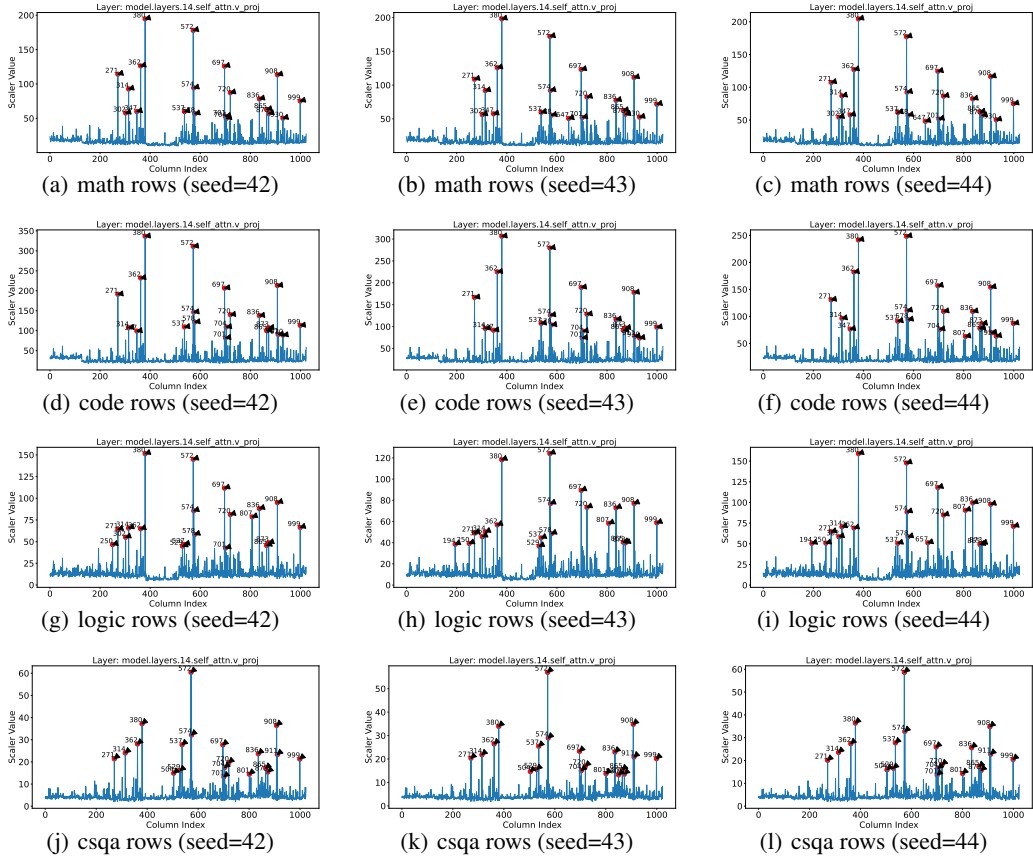

Figure 13: Delta-scale rows of `model.layer.14.self_attn.v_proj` with distinct data samples on different reasoning Llama3-8B models.

In Mistral-7B and Qwen2.5-14B, the patterns are also like in Llama3-8B, we visualize the `model.layer.24.mlp.gate` for each reasoning data in Figures 11 and 12. We can observe that math and code abilities share a large proportion of common DSR parameters, while others do not, such a phenomenon reminds us that the benefits and conflicts are entangled. Therefore, we can see the math and code performances of Mistral-7B and Qwen2.5-14B in Table 2 and Table 15 are in strong correlation, which can also align with the finding in Section 3.2.

Classical meta-learning methods like MAML/Reptile learn *how to adapt*: they optimize an initialization or update rule so that a few gradient steps on a *new* task lead to good performance, typically via an explicit meta-objective and episodic training [1-2]. In contrast, DiFT assumes a fixed set of tasks and uses *standard SFT*; DSR is a post-hoc, row-wise saliency analysis that reorganizes where updates happened without the additional meta-optimization loop. Conceptually, meta-learning aims to achieve **fast adaptation**, whereas our DiFT focuses on **reducing interference**. These two paradigms can be complementary: a meta-learning initialization could still be optimized by DiFT.

MoE architectures (e.g., GShard, Switch Transformer) introduce explicit experts plus a learned router, enabling conditional computation but requiring architectural changes, routing stability tricks, and specialized infra [3-4]. DiFT instead treats a dense LLM as containing *implicit experts*, i.e., DSR-identified subsets, and enforces specialization via freezing, with unchanged architecture and inference cost.

We indeed point out that unstable routing can complicate the direct application of our method in the Limiations. We would like to clarify that:

- Our concern with MoE is primarily about the instability of expert routing during training, more stable parameter subsets need more data to locate, which makes the notion of a stable

task-specific parameter subset harder to define and track. This is a practical challenge, rather than a theoretical incompatibility between our method and MoE architectures.

- We see our current dense-model study as a foundation. Understanding and validating the method in a simpler, more stable dense setting is a natural first step before tackling the additional complexity introduced by dynamic routing in MoE.

