# OpenReview forum: "Differential Fine-Tuning Large Language Models Towards Better Diverse Reasoning Abilities"
_ICLR.cc/2026/Conference — ICLR 2026 Poster_

### Official Review · Reviewer_kiFD · 2025-10-29

**Soundness:** 3
**Presentation:** 3
**Contribution:** 2
**Rating:** 4
**Confidence:** 2

**Summary:**

This paper proposes Differential Fine-Tuning (DiFT), a novel fine-tuning strategy designed to mitigate task conflicts while preserving performance benefits across multiple reasoning tasks in large language models (LLMs). The method identifies task-specific parameters through a Delta-Scale Row (DSR) analysis and selectively updates them to avoid destructive interference among reasoning abilities. Experimental results on Llama3-8B, Mistral-7B, and Qwen2.5-14B demonstrate that DiFT can outperform existing mix-up and continual fine-tuning baselines such as DMT, CoBa, HFT, and LoTA, improving average target accuracy (ATA) across various reasoning benchmarks including GSM8k, xGLUE, LogiQA2, and CSQA. The work is well-motivated and offers an interesting parametric perspective on reasoning conflicts in LLM fine-tuning.

**Strengths:**

1.The paper addresses an important and relatively underexplored problem—task interference and reasoning conflicts during supervised fine-tuning of LLMs.
2.The DSR-based parameter analysis provides an insightful method to identify task-sensitive parameters, which could inspire future research on interpretable fine-tuning.
3.The proposed DiFT framework is conceptually elegant and achieves consistent empirical gains across diverse reasoning benchmarks and model sizes.
4.The authors also provide comprehensive experiments, ablation studies, and discussions comparing mix-up and continual learning paradigms, which enhance the credibility of their findings.

**Weaknesses:**

While I find the idea of DiFT quite promising, I have several concerns and questions that the authors may wish to address to strengthen the work:
1.Scalability with Task Number. I express concerns about the scalability of the proposed DiFT method, noting that maintaining task-specific parameter updates for multiple reasoning tasks may cause the model size and memory usage to grow as the number of tasks increases. It is therefore recommended to conduct experiments with different numbers of tasks (e.g., two, three, or more) to evaluate how DiFT performs in large-scale and multi-domain settings. Such experiments would help demonstrate the method’s practicality and efficiency in real-world applications involving large models.

2.Relation to Meta-Learning and Mixture-of-Experts Paradigms. I am interested in how DiFT conceptually and empirically compares with existing meta-learning and multi-expert approaches that also address knowledge sharing and task specialization across multiple domains. It is suggested that the paper explicitly discuss these connections, highlighting the key distinctions between DiFT and meta-learning or expert-based modular frameworks. Additionally, including a comparative performance analysis—at least at a conceptual level—would help clarify the unique contribution and positioning of DiFT within the broader landscape of multi-task learning methods.

3.Substitution of the Key Parameter Identification Mechanism. The DSR mechanism plays a central role in DiFT, but I wonder how it compares with other possible parameter importance estimation techniques, such as attention-based attribution or gradient sensitivity analysis. Could the authors provide either an ablation about substituting DSR with attention-based mechanisms to examine robustness and generality of the framework?

4.Limited Experimentation on Larger Models. The authors mention that, “Due to hardware limitations, we only conducted experiments on 7/8B and 14B LLMs in this paper, lacking validation on larger-scale (30B+) models that can be complementary.” However, larger-scale validation seems critical to the paper’s claim that DiFT improves diverse reasoning capabilities of LLMs in general. Since scalability is a core aspect of language model research, I strongly recommend that the author provide results on the expected performance of DiFT on various different-sized models.  Without this, it remains unclear whether the proposed approach would maintain its efficiency and stability at scale.

5.Lack of Convergence Analysis in Fine-Tuning. The paper does not provide any theoretical or empirical evidence of convergence during fine-tuning. Since the proposed DiFT method involves differential parameter updating across multiple reasoning tasks, it is unclear whether the optimization process is guaranteed to converge to a stable point. Moreover, when the number of tasks increases, convergence behavior could vary substantially across different parameter subsets. I am particularly concerned that without monitoring the fine-tuning loss or parameter change trajectories, it is difficult to ensure that each reasoning sub-model (or the overall DiFT model) has reached a stable equilibrium. Providing convergence curves for varying task counts (e.g., 2, 3, or more reasoning datasets) would significantly strengthen the technical soundness of this work.

6.Missing Computational Cost and Efficiency Analysis. The paper does not provide sufficient information about how computational efficiency scales with the number of fine-tuning tasks. While Section 5.1 briefly mentions that the computing costs are lighter than task-vector and gradient-based methods, it lacks quantitative evidence showing how training time, token consumption, and GPU (CUDA) utilization change as the number of reasoning tasks increases. Such data are crucial to assess whether DiFT remains efficient and feasible when extended to larger task sets. Including analyses of runtime, token usage, and GPU memory consumption under different task counts would help clarify the scalability and practical efficiency of the proposed method compared to baselines like LoTA or HFT.

**Questions:**

See weaknesses

---

> ### Author Response · Authors · 2025-11-21
> **Reply (1/3) to W1-W2**
>
> Thanks a lot for your time and efforts on reviewing our submitted manuscript, and we are really grateful for valuable reviews! The following contents are our responses to the raised weaknesses, questions, and suggestions.
>
> ### **W1. Scalability with Task Number.**
> Thank you for the valuable suggestion! Our research focuses on diverse reasoning capabilities, and commone reasoning tasks are math, code, logic, commonsense, which have been incorporated in our experiments. In the original manuscript, we evaluate the proposed DiFT on 2 and 3 reasoning tasks, demonstrating the consistent effectiveness. Indeed, more tasks can enhance the scalability of the DiFT, so we perform 4 -task and 5-task experiments, results are as below:
> | Llama3-8B | GSM8k | xGLUE | LogiQA2 | CSQA | HotPotQA |
> | -- | -- | -- | -- | -- | -- |
> | mix-math-code-logic-csqa | 67.93 | 1.0058 | 31.55 | 77.07 | - |
> | +DiFT | **69.22** | **1.1342** | **33.21** | **77.31** | - |
> | mix-math-code-logic-csqa-hotpotqa | 62.62 | 0.9971 | 33.14 | 77.97 | 36.74 |
> | + DiFT | **62.68** | **1.0413** | **33.84** | **79.36** | **38.45** |
>
> Empirical results of 4-task and 5-task mix-up DiFT illustrate consistent improvements, demonstrating the effectiveness and more tasks potential of DiFT.
>
> We notice that distinct tasks have different trainable parameter scales, we statistic the overlap rate of DSR parameters between any newly added task and historical tasks with the task number (K) increasing:
> | Task Num | Task | overlap rate (%) | param ratio (%) |
> | -------- | ------- | ---------------- | ------ |
> | K=1 | math | — | 3.7 % |
> | K=2 | code | — | 5.8 % |
> | K=3 | math | 54.3 | 7.2 % |
> | | code | 52.5 | |
> | | logic | 46.8 | |
> | K=4 | math | 63.8 | 8.6 % |
> | | code | 57.2 | |
> | | logic | 53.5 | |
> | | csqa | 63.4 | |
> | K=5 | math | 70.5 | 10.5 % |
> | | code | 67.2 | |
> | | logic | 68.9 | |
> | | csqa | 78.6 | |
> | | hotpoqa | 41.4 | |
>
> From this table, we can notice that with the task number increasing, each single task has more overlapped DSR parameters, thereby the growth of total trainable parameters slows down, showing a convergent trend. Such results indicate that the trainable parameter scale cannot easily as the same as full SFT, as extensive reasoning tasks can be decomposed into fundamental ones, and mix-up parameter union cannot expand very fast.
>
> **We have highlighted these experiments in Section 5.2, and added the 4-task and 5-task experiments, and the statistical visualizations in the updated manuscript's Appendix F.5.**
>
> ### **W2. Relation to Meta-Learning and MoE architecture.**
> You raised a great question! And we would like to clarify the relations between the proposed DiFT and Meta-learning/MoE models.
>
> **Compare to meta-learning.** Classical meta-learning methods such as MAML/Reptile learn *how to adapt*: they optimize an initialization or update rule so that a few gradient steps on a *new* task lead to good performance, typically via an explicit meta-objective and episodic training [1-2]. In contrast, DiFT assumes a fixed set of reasoning tasks and uses *standard SFT*; DSR is a post-hoc, row-wise saliency analysis that reorganizes where updates happened, without the additional meta-optimization loop. Conceptually, meta-learning aims to achieve **fast adaptation**, whereas our DiFT focuses on **reducing interference**. These two paradigms can be complementary: a meta-learning initialization could still be optimized by DiFT.
>
> **Compare to MoE / multi-expert frameworks.** MoE architectures (e.g., GShard, Switch Transformer) introduce explicit experts plus a learned router, enabling conditional computation but requiring architectural changes, routing stability tricks, and specialized infra [3-4].  DiFT instead treats a dense transformer as containing *implicit experts* (DSR-identified subsets) and enforces specialization via masking, with unchanged architecture and inference cost.
>
> We have clarified these connections in the paper and position DiFT as:
> * **Orthogonal to meta-learning**, while the mix-up and continual SFT across distinct reasoning tasks have no meta-objective, same SFT loop;
> * **Architecturally lighter than MoE**, while empirically achieving multi-domain specialization within a single dense model.
>
> **We have highlighted these contents in Section 4.1 and added this discussion at the end of Appendix F.6 of the revised manuscript.**

---

> ### Author Response · Authors · 2025-11-21
> **Reply (2/3) to W3-W4**
>
> ### **W3. Substitution of the Key Parameter Identification Mechanism.**
> Your suggestion is insightful! To evaluate the DiFT performance in the continual SFT setting, we select HFT and LoTA as the continual SFT baselines given their superiority. Your suggestion is valuable, the EWC is classic and L2-SP is theoretically well-designed, and can be applied in LLMs. We supplement the continual SFT experiments based on these 2 baselines, results are as follows:
> | Llama3-8B | GSM8k | xGLUE | LogiQA2 |
> | --- | --- | --- | --- |
> | seq-math-code | 44.35 | 0.9902 | 32.83 |
> | +EWC | 43.98 | 0.9850 | 32.50 |
> | +L2-SP | 44.12 | 0.9880 | 32.65 |
> | +DiFT | **46.32** | **1.0557** | 35.86 |
> | seq-math-logic | 10.99 | 0.6433 | 31.30 |
> | +EWC | 10.85 | 0.6380 | 31.10 |
> | +L2-SP | 10.92 | 0.6410 | 31.20 |
> | +DiFT | **11.37** | 0.6919 | **31.23** |
>
> EWC and L2-SP are both classic baselines, but not strong enough as HFT and LoTA in our baselines. Through these experiments, we can affirm the superiority of the proposed DiFT.
>
> **We have highlighted these contents in Section 5.1 and supplemented this experiments in Appendix F.4 of the revised manuscript.**
>
> ### **W4. Limited Experimentation on Larger Models and MoE.**
> In the initial manuscript, we verify the proposed DiFT on 7B to 14B models, including Llama‑3, Mistral, and Qwen‑2.5. Due to that full SFT with 32B LLMs demands more A100 GPUs, as we mentioned in the **Limitation section**, we have not enough computing resources to conduct 32B experiments for now. We promise once obtaining enough computing resources, and we have strengthen this statement in the Limitation section.
>
> Even we have such resource limitations, to further validate the scalability of our strategy, we conduct experiments on Qwen2.5-3B, and results are as follows:
> | Qwen2.5-3B | GSM8k | xGLUE | LogiQA2 |
> | -- | -- | -- | -- |
> | mix-math-code | 68.01 | 1.2102 | 35.69 |
> | +DiFT | **69.83** | **1.2213** | 35.94 |
> | mix-code-logic | 55.12 | 1.1997 | 36.90 |
> | +DiFT | 61.87 | **1.2375** | **37.10** |
> | seq-math-code | 60.05 | 1.1910 | 35.50 |
> | +DiFT | **61.79** | **1.1924** | 35.20 |
> | seq-math-logic | 62.93 | 0.9872 | 36.70 |
> | +DiFT | **64.37** | 0.9912 | **37.60** |
>
> We can see that DiFT can also work very well on Qwen2.5-3B on both mix-up and continual SFT settings. Despite the computing limitation, our experiments on 3B models, in conjunction with results from 7B to 14B models, consistently demonstrate the effectiveness of our method, suggesting its scalability with increasing model size (3B->7B->8B->14B). The DiFT strategy relies solely on DSR parameter selection, and larger models exhibit fundamentally similar DSR parameter nature.
>
> Furthermore, validating the consistent improvements of our method on models of this scale is particularly meaningful, as these model sizes are commonly employed in real-world deployments. **We also plot figures to illustrate that DiFT can be scalable and consistent across different sizes of LLMs.**
>
> **We have highlighted the scalability in Section 5, and included the Qwen2.5-the 3B experiments and scaling visualizations in Appendix E in the updated manuscript.**
>
> **On MoE architectures and routing stability Regarding MoE models.** We indeed point out that unstable routing can complicate the direct application of our method in the Limitations. We would like to clarify additional 2 points:
>
> a. **Current limitation is practical, not conceptual.** Our concern with MoE is primarily about the instability of expert routing during training, which makes the notion of a stable task-specific parameter subset harder to define and track. This is a practical challenge, rather than a theoretical incompatibility between our method and MoE architectures.
>
> b. **A foundation for future MoE extensions.** We see our current dense-model study as a foundation. Understanding and validating the method in a simpler, more stable dense setting is a natural first step before tackling the additional complexity introduced by dynamic routing in MoE.
>
> **We have highlighted related contents in Section 4.1 and supplemented more MoE-related discussion at the end of Appendix F.6 of the revised manuscript.**

---

> ### Author Response · Authors · 2025-11-21
> **Reply (3/3) to W5-W6**
>
> ### **W5. Lack of Convergence Analysis in Fine-Tuning.**
> We introduce our detailed experimental settings in Appendix B.1, and our SFT hyperparameters can guarantee the training process is convergent. However, your suggestion is valuable, and we have now added some training loss figures to the updated manuscript to illustrate that.
>
> We introduce our detailed experimental settings in Appendix B.1, and our SFT hyperparameters can guarantee the training process is convergent. However, your suggestion is valuable, and we have now **added training loss figures to the Appendix B.1 updated manuscript** to illustrate that.
>
> ### **W6. Missing Computational Cost and Efficiency Analysis.**
> In Appendix B.2, we introduce the costs of the DSR computing, which is very computing-friendly "probing" stage compared to the main experiments. The "probe SFT" is indeed an extra step compared to vanilla SFT, however, such a step introduce quite limited resources: a 1k (1/20 than full) Llama3-8B SFT experiment consumes merely 1/20 time costs (6 minutes on 8xA100), and inference on the same serve needs less than 2 minutes. Therefore, the "probe" stage introduces 8 minutes, given the full SFT on the same device demands 120 minutes, which is 15 times of the "probe" step, the "probe" complexity is marginal.
>
> With the extra 1/15 cost, we can **produce an efficient SFT of merely 3%-5% trainable parameters, saving much time and computing than the full SFT**. More importantly, our DiFT strategy provides **better interpretability** by locating task-specific DSR parameters, and achieve **superior diverse reasoning performance** through mitigating conflicts in vanilla SFT.
>
> **We have supplemented the computing costs in the augmented Appendix B.3 of the updated manuscript.**
>
> **References:**
>
> [1] Model-Agnostic Meta-Learning for Fast Adaptation of Deep Networks. ICML 2017.
>
> [2] On First-Order Meta-Learning Algorithms. arXiv:1803.02999.
>
> [3] GShard: Scaling Giant Models with Conditional Computation and Automatic Sharding. ICLR 2021.
>
> [4] Switch Transformers: Scaling to Trillion Parameter Models with Simple and Efficient Sparsity. JMLR 2022.

---

### Official Review · Reviewer_os6g · 2025-11-01

**Soundness:** 3
**Presentation:** 3
**Contribution:** 2
**Rating:** 4
**Confidence:** 3

**Summary:**

This paper investigates the complex problem of task conflict and synergy during LLM fine-tuning. The authors observe that training a model on one reasoning task can often degrade its performance on another. To address this, they introduce a parameter identification method called Delta-Scale Row (DSR) to find the parameters most critical to a specific task. They then propose a new fine-tuning strategy, Differential Fine-Tuning (DiFT), which selectively updates or freezes these DSR parameters based on the learning scenario. The goal is to mitigate the conflicts that arise when mixing tasks or learning them sequentially. The authors provide experimental results across several models (Llama3, Mistral, Qwen2.5) to demonstrate that DiFT outperforms existing baseline methods.

**Strengths:**

Clear writing. The motivation, pipeline, and training regimes are easy to follow. Key concepts (DSR, union/difference rules) are defined and used consistently.


New method. DiFT introduces a simple activation-delta criterion to pick “sensitive rows,” then applies union/difference rules for mix-up and continual learning. The idea is lightweight and practical.


Extensive experiments. Results span multiple models and task families, with ablations and sanity checks (e.g., inverse variants). Findings appear consistent across settings.

**Weaknesses:**

(1) Assumption validation is improved but still incomplete.
Inverse-DiFT and random-selection baselines support necessity and non-randomness. Still missing stronger causal probes. Cross-task “row transplant” tests, DSR computed on disjoint sources, and counterfactual swaps across tasks would better establish causality beyond necessity.

(2) Heuristic choices lack broader justification.
The ablation on the number of rows (C) addresses set size. However, the union rule (mix-up) and the difference rule (continual) remain untested against plausible alternatives. No intersection, weighted unions, or trust-region style variants. No analysis of when overlap helps vs. harms. Results could benefit from a principled rationale or a small design sweep in the main text.

(3) Missing continual-learning baselines.
No direct comparisons with parameter-importance methods: EWC/Online-EWC, L2-SP, MAS, SI, PackNet, LwF. Without them, placement within the CL literature is unclear.

(4) Potential signal–target entanglement.
DSR selection uses the same task family that later benefits in training/evaluation. This can inflate ATA via selection–target coupling. Need results where DSR is computed on held-out variants or adjacent corpora.

**Questions:**

(1) Signal–target entanglement.
DSR is computed with data from the target task, then those rows are trained on mixtures including the same family. Could ATA gains partially reflect selection–target leakage?

(2) Relation to importance metrics.
How does DSR compare against Fisher information, diagonal Hessian, or gradient-based importance (EWC, L2-SP, PackNet, LwF) under identical budgets?

(3) Layer and granularity choices.
Which layers are hooked—every linear layer or a subset? How sensitive are results to selecting rows in early vs. deep layers?

(4) Continual rule clarity.
For DSR_diff, does training freeze everything except the newly identified rows, or only historical DSR sets? What remains trainable for rows never selected by any task? Please show capacity remaining over steps (k).

---

> ### Author Response · Authors · 2025-11-21
> **Reply (1/2) to W1-W4, Q1**
>
> Thank you for your devoting to reviewing our paper, and we are sincerely grateful for the valuable reviews, the following contents are our responses to your raised weaknesses and questions.
>
> ### **W1. DSR causality beyond necessity, add cross-task “row transplant” to validate casality.**
> To go beyond inverse/random-DiFT ablations, we perform a **cross-task* DSR transplant* between the math-tuned and csqa-tuned Llama3-8B models. We copy only the top-100 DSR params for math into the csqa-tuned model (csqa-sub-math), and symmetrically, the csqa DSR params into math-tuned model, results are as follows:
> | model | GSM8k | CSQA |
> | -- | -- | -- |
> | math-sft | 61.64 | 67.24 |
> | csqa-sft | 8.79 | 79.36 |
> | math-sub-csqa | 41.02 | 75.72 |
> | csqa-sub-math | 22.21 | 71.48 |
>
> We can observe that injecting csqa-DSR into math degrades GSM8k while improving CSQA, and replacing csqa model with math-DSR enhances GSM8k but hurts CSQA. Such results constitute a causal intervention: moving DSR params diverts the correponding reasoning ability and introduces interference when transplanting into other tasks.
>
> **We have highlighted that in Section 4.1, and added the cross-task transplant experiments in Appendix D.2 to strengthen the causality of DSR parameters in the revised manuscript.**
>
>
> ### **W2. Heuristic choices lack broader justification.**
> Your suggestion is valuable! To test whether our union rule is an arbitrary heuristic, we perform a **weighted-union variant**: starting from the Llama-3-8B-Base, we scale rows in the math-logic DSR union by 1.5 and all other rows by 0.9 ("scaled-base-dsr-union-math-logic").
> | model | GSM8k | xGLUE | LogiQA2 | CSQA |
> | -- | -- | -- | -- | -- |
> | Llama3-8B | 39.42 | 1.0874 | 31.93 | 69.29 |
> | vanilla-math-sft | 61.64 | 1.2228 | 30.73 | 67.24 |
> | vanilla-logic-sft | 30.17 | 0.6880 | 37.02 | 72.89 |
> | vanilla-mix-sft-math-logic | 64.37 | 1.2092 | 32.32 | 70.52 |
> | scaled-base-dsr-union-math-logic | 42.30 | 1.0918 | 32.38 | 69.53 |
>
> In this table, the simple re-weighting already shifts ability toward math/logic (GSM8k 39.42->42.30, LogiQA2 31.93->32.38) with only minor changes on xGLUE/CSQA, confirming that the DSR union indeed concentrates task-relevant signal. Moreover, the performance remains between vanilla base and full mix-SFT, indicating that the union strategy is a stable rather than a simple heuristic choice.
>
> **We have highlighted relalted contents in Section 4.1 and added the scaled-union experiments in Appendix D.2 to strengthen the causality of DSR.**
>
>
> ### **W3. Missing continual-learning baselines.**
> To evaluate the DiFT performance in the continual SFT setting, we select HFT and LoTA as the continual SFT baselines given their superiority. Your suggestion is valuable, the EWC is classic and L2-SP is theoretically well-designed, and can be applied in LLMs. We supplement the continual SFT experiments based on these 2 baselines, results are as follows:
> | Llama3-8B | GSM8k | xGLUE | LogiQA2 |
> | --- | --- | --- | --- |
> | seq-math-code | 44.35 | 0.9902 | 32.83 |
> | +EWC | 43.98 | 0.9850 | 32.50 |
> | +L2-SP | 44.12 | 0.9880 | 32.65 |
> | +DiFT | **46.32** | **1.0557** | 35.86 |
> | seq-math-logic | 10.99 | 0.6433 | 31.30 |
> | +EWC | 10.85 | 0.6380 | 31.10 |
> | +L2-SP | 10.92 | 0.6410 | 31.20 |
> | +DiFT | **11.37** | 0.6919 | **31.23** |
>
> EWC and L2-SP are both classic baselines, but not strong enough as HFT and LoTA in our baselines. Through these experiments, we can affirm the superiority of the proposed DiFT.
>
> **We have highlighted these contents in Section 5.1, and supplemented this experiments in Appendix F.4 of the revised manuscript.**
>
> ### **W4 & Q1. Potential signal–target entanglement.**
> We have fine-tuned the base LLM with 3 couples of 1k math samples and a full SFT with 20k math data, which are of no overlapped data, and use the same inference data to compute DSR of these finetuned LLMs. These math-finetuned LLMs share majority DSR parameters as shown in the following table:
> | overlap ratio (%) | math-1k-42 | math-1k-43 | math-1k-43 | math-20k |
> | --- | --- | --- | --- | --- |
> | math-1k-42 | 100 | 97.42 | 98.75 | 98.36 |
> | math-1k-43 | 97.42 | 100 | 98.07 | 97.92 |
> | math-1k-44 | 98.75 | 98.07 | 100 | 99.04 |
> | math-20k | 98.36 | 97.92 | 99.04 | 100 |
>
> We can observe that all the math training data exhibit high consistency DSR parameters, which is in line with the derivations in Section 3.2 and the empirical phenomena in Section 3.3.

---

> ### Author Response · Authors · 2025-11-21
> **Reply (2/2) to Q2-Q4**
>
> ### **Q2. Relation to other importance metrics.**
> Let $P$ be the parameter scale, $B$ be training data scale per task in full SFT, $N$ be the task number, and let a forward / backward pass each cost $O(P)$ FLOPs/token.
>
> **our DiFT pipeline**
> * **Pre-SFT:** We only train on a *small probe subset* $B_{\text{probe}}\ll B$ (1k vs 20k), which empirically yields almost identical top-DSR sets as full SFT. The *Complexity* is: $O(N,B_{\text{probe}},P)$.
> * **DSR scoring:** For each task, we run **forward-only** passes of base vs. probe model on tiny calibration sets (5 groups of 50 samples) and accumulate row-wise squared activation differences.  This is $O(N,M,P)$ w. $M\ll B$, which is negligible compared to multi-GPU SFT.
> * **DiFT training:** Fintuning only a small scale of DSR params compared to standard SFT. *Complexity* is $O(N,B,P)$, i.e., equal to standard SFT down to a $< 1.0$ constant.
> Total overhead vs. vanilla SFT is thus $O\big(N(B_{\text{probe}}+M)P\big)$ with $B_{\text{probe}},M\ll B$; in our setting this is only a few percent of the full SFT FLOPs.
>
> **Fisher / diagonal Hessian / EWC**
> * **Diagonal Fisher / EWC.** To estimate diagonal Fisher, EWC scans the task data and accumulates squared gradients for each parameter, requiring an *extra backward pass* over the Fisher dataset. Complexity of each task is $O(B_{\text{Fisher}},P)$ with a full backward, typically $B_{\text{Fisher}}\approx B$. So the overhead is on the order of **another full epoch of SFT**: $O(N,B,P)$, substantially larger than our $O(N(B_{\text{probe}}+M)P)$. Memory also stores a full diagonal Fisher vector of size $O(P)$.
> * **Diagonal Hessian.** Diagonal-Hessian methods approximate curvature via Hessian–vector products or Hutchinson-style probes, each costing $\approx 2$ gradient passes. With $K$ probes, this is $O(K,B,P)$ FLOPs per task $(K\ge 1)$, i.e., *multiples* of EWC's cost and far above our small forward-only DSR pass.
>
> **Other gradient-based importance schemes**
> * **PackNet.** Requires *iterative pruning + retraining* cycles for each task: train - prune - retrain, potentially repeated several times, yielding $\approx O(J,B,P)$ with $J>1$ effective full passes, which is much heavier than our forward analysis.
> * **LwF.** For each mini-batch, LwF requires a *teacher model* forward alongside the student and compute a distillation loss. Complexity per step is roughly doubled on the forward side: $O(B,P)$ extra FLOPs per batch over the full training trajectory.
>
> In summary, the computing complexity of our DiFT and other parameter importance methods are as the table below:
> | Algorithm | Complexity |
> |------------|---------------|
> | vanilla SFT | $O(N,B,P)$ |
> | DSR / DiFT | $O(N (B_{probe} + M) P)$, with $B_{probe}$, M $\ll$ B |
> | EWC / diag Fisher | $O(N, B, P)$ via extra backward passes |
> | Diagonal Hessian methods | $O(N, K, B, P)$, $K > 1$ |
> | PackNet / LwF | At least one extra SFT-scale pass or an extra forward per step |
>
> Thus, for LLM-scale models, DSR offers **importance estimates at a fraction of the FLOPs** required by Fisher/Hessian-based or distillation-based baselines, while keeping the training loop as simple as vanilla SFT.
>
> **We have highlighted that in Section 5.1 and included the computing complexity comparisons in Appendix B.2 of the revised manuscript.**
>
> ### **Q3. Layer and granularity choices.**
> As we introduce in section 3.2, we compute to obtain DSR across all named parameters of given LLMs. In our implementaion (in the supplmentary materials), we register forward hook on the same type of **every transformer block**, and calculate DSR "per row, per layer", then DiFT selects top-C rows "per layer". Thus, the DiFT based on DSR is "layer-agnostic".
>
> ### **Q4. What remains trainable for rows never selected by any task?**
> In section 4.3, Eq.(8) defines the continual SFT rule of DiFT, where each time we fine-tune the DSR of the current task by subtracting historical tasks' DSR to maintain the learned reasoning abilities. We notice that distinct tasks have quite different trainable parameter scales, during the continual SFT setting Math-Code-Logicot-CSQA, the trainable parameter ratio (compared to the entire LLM) fluctuates as follows:
> | task | param ratio (%) |
> | --- | --- |
> | math | 3.7 |
> | code | 1.4 |
> | logicot | 1.9 |
> | csqa | 2.4 |
>
> We can see that the trainable parameter scale depends more on the interactions among tasks. For csqa whose DSR is largely different from historical tasks, its capacity for active parameters is almost equal to its DSR.

---

### Official Review · Reviewer_cicY · 2025-11-03

**Soundness:** 2
**Presentation:** 3
**Contribution:** 3
**Rating:** 6
**Confidence:** 2

**Summary:**

The paper introduces **Differential Supervised Fine-Tuning (DiFT)**, a principled framework to mitigate conflicts among multiple reasoning tasks during supervised fine-tuning (SFT) of large language models (LLMs).
The authors observe that when LLMs are fine-tuned jointly or sequentially on heterogeneous reasoning datasets (e.g., math, code, logic, commonsense), performance may degrade due to destructive interference. Through comprehensive experiments on Llama3-8B, Mistral-7B, and Qwen2.5-14B, they show that task combinations can yield both mutual benefits and conflicts, motivating a more nuanced SFT approach.

The paper’s central insight is that each reasoning ability corresponds to a subset of model parameters that are disproportionately influential for that task. To identify these, the authors propose a quantitative metric called the Delta-Scale Row (DSR) score, computed from activation differences between base and fine-tuned models.
Building upon DSR analysis, DiFT differentially updates parameters. This selective parameter updating reduces interference and preserves mutual benefits across tasks.
Empirically, DiFT achieves consistent improvements across math (GSM8k), code (xGLUE), logic (LogiQA2), and commonsense (CSQA) reasoning benchmarks, outperforming strong baselines such as HFT, LoTA, DMT, and CoBa. The proposed analysis and method generalize across both base and instruct LLMs, showing broad applicability.

**Strengths:**

1. The Delta-Scale Row (DSR) analysis offers a novel and interpretable metric for identifying task-specific sensitive parameters, bridging parameter variation analysis with reasoning-oriented fine-tuning.

2. The discovery of reasoning task interactions—synergistic or conflicting—across math, code, logic, and commonsense reasoning provides valuable empirical insight into multi-ability LLM training.

3. The theoretical motivation for DSR is grounded in Taylor expansion and sensitivity analysis, giving analytical meaning to the empirical findings.

4. Contributes to a deeper understanding of task interference and transfer in multi-objective fine-tuning, a core challenge for multi-ability LLMs.

**Weaknesses:**

1. While DSR analysis is well-motivated, the connection between row-wise sensitivity and global reasoning dynamics remains empirical. A theoretical characterization of why DSR partitions correlate with reasoning abilities would strengthen the contribution.
2. Although the paper claims DiFT is computationally efficient, quantitative runtime and FLOPs comparisons with baselines are absent; adding explicit cost metrics would improve clarity.
3. Experiments focus on four reasoning tasks; extending to non-symbolic reasoning (e.g., spatial or multi-hop tasks) would further validate generality.
4. Although results generalize to different LLMs, scaling behavior (e.g., how DSR sparsity changes with model size) is not analyzed.

**Questions:**

1. How stable are DSR distributions across random seeds or dataset subsamples? Does the identified task-specific subspace remain consistent under noisy data?

2. Can DSR be estimated during training rather than post-hoc, enabling online conflict detection and adaptive freezing?

3.  How sensitive is DiFT to the hyperparameter controlling the top-$C$ DSR threshold? Are results robust to different $C$ values?

---

> ### Author Response · Authors · 2025-11-21
> **Reply (1/3) to W1**
>
> ### **W1. A theoretical characterization of DSR partitions correlate with reasoning abilities.**
> In Section 3.2, we derived that the inference-time differences (i.e., DSR) between finetuned-base LLMs are the results of integreted optimization for parameters by gradients, and empirical results illustrated the reasoning task differences in Section 3.3. However, your suggestion of theoretical explanation for distinct parameter subsets can be a great complementary for our work.
>
> Our DSR "sensitivity" is defined **per row** $k$ of a named parameter as $s _{k} = \mathbb{E} _{t}||\Delta Y ^{k} _{t}|| _{2} ^{2}$, the expected change in that row's output after task-oriented SFT. For a linear row $Y ^{k} _{t}=W _{k}X_t$, $\Delta Y ^{k} _{t} = (W _{k}^{\text{ft}}-W _{k}^{\text{base}})X _t = \Delta W _k X _t.$ If input statistics are approximately fixed during small-step SGD, $s _k \approx ||\Delta W _{k}|| _{2} ^{2} \cdot \mathbb{E} _t||X _t|| _2^2$, DSR is thus proportional to the **squared parameter change of row (k)**. With SGD, $\Delta W _k = -\eta\sum _t g ^{k} _{t}$, giving $\mathbb{E}||\Delta W _k|| ^{2} _{2} \propto \sum _t \mathbb{E}||g ^{k} _{t}|| _{2} ^{2}$, i.e., **integrated gradient energy**. For log-likelihood optimized models, $\mathbb{E}||g _k|| _2^2$ equals the diagonal of the **Fisher Information Matrix**, the standard notion of parameter importance used in EWC [1], SI [2], MAS [3], etc [4,5,6,7]. Thus DSR is a **row-granular Fisher/importance estimator**: rows with large DSR are exactly those repeatedly used to reduce the task's loss, which explains why their partitions correlate with specific reasoning abilities.
>
> We use **rows**, instead of columns or individual elements, because a row $W _{k}$ defines one neuron/channel $y _{k}=W _{k}x$, which is a natural *functional unit* that downstream non-linear and mixing treat as a coherent feature or circuit[8]. Columns mix into all neurons and single weights are not meaningful units. Row-wise aggregation also reduces variance versus element-wise scores and yields clean, structured masks (channel-level sparsity) that are easy to implement. So row-wise DSR combines a clear Fisher-based theory with stable, interpretable, and practical units of specialization.
>
> **We have highlighted these in Section 4.1 and updated these interpretations in the Appendix D.1 of the revised manuscript.**

---

> ### Author Response · Authors · 2025-11-21
> **Reply (2/3) to W2**
>
> ### **W2. DiFT computationally complexity.**
> In Appendix B.2, we introduce DSR computing costs, which is very limited compared to full SFT. Let $P$ be the parameter scale, $B$ be training data scale per task in full SFT, $N$ be the task number, and let a forward / backward pass each cost $O(P)$ FLOPs/token.
>
> **our DiFT pipeline**
>
> * **Pre-SFT:** We only train on a *small probe subset* $B _{\text{probe}}\ll B$ (1k vs 20k), which empirically yields almost identical top-DSR sets as full SFT. The *Complexity* is: $O(N,B _{\text{probe}},P)$.
> * **DSR scoring:** For each task, we run **forward-only** passes of base vs. probe model on tiny calibration sets (5 groups of 50 samples) and accumulate row-wise squared activation differences.  This is $O(N,M,P)$ w. $M\ll B$, which is negligible compared to multi-GPU SFT.
> * **DiFT training:** Fintuning only a small scale of DSR params compared to standard SFT. *Complexity* is $O(N,B,P)$, i.e., equal to standard SFT down to a $< 1.0$ constant.
> Total overhead vs. vanilla SFT is thus $O\big(N(B _{\text{probe}}+M)P\big)$ with $B _{\text{probe}},M\ll B$; in our setting this is only a few percent of the full SFT FLOPs.
>
> **Fisher / diagonal Hessian / EWC**
> * **Diagonal Fisher / EWC.** To estimate diagonal Fisher, EWC scans the task data and accumulates squared gradients for each parameter, requiring an *extra backward pass* over the Fisher dataset. Complexity of each task is $O(B _{\text{Fisher}},P)$ with a full backward, typically $B _{\text{Fisher}}\approx B$. So the overhead is on the order of **another full epoch of SFT**: $O(N,B,P)$, substantially larger than our $O(N(B _{\text{probe}}+M)P)$. Memory also stores a full diagonal Fisher vector of size $O(P)$.
> * **Diagonal Hessian.** Diagonal-Hessian methods approximate curvature via Hessian–vector products or Hutchinson-style probes, each costing $\approx 2$ gradient passes. With $K$ probes, this is $O(K,B,P)$ FLOPs per task $(K\ge 1)$, i.e., *multiples* of EWC's cost and far above our small forward-only DSR pass.
>
> **Other gradient-based importance schemes**
> * **PackNet.** Requires *iterative pruning + retraining* cycles for each task: train - prune - retrain, potentially repeated several times, yielding $\approx O(J,B,P)$ with $J>1$ effective full passes, which is much heavier than our forward analysis.
> * **LwF.** For each mini-batch, LwF requires a *teacher model* forward alongside the student and compute a distillation loss. Complexity per step is roughly doubled on the forward side: $O(B,P)$ extra FLOPs per batch over the full training trajectory.
>
> In summary, the computing complexity of our DiFT and other parameter importance methods are as the table below:
> | Algorithm | Complexity |
> |------------|---------------|
> | vanilla SFT | $O(N,B,P)$ |
> | DSR / DiFT | $O(N (B_{probe} + M) P)$, with B_probe, M $\ll$ B |
> | EWC / diag Fisher | $O(N, B, P)$ via extra backward passes |
> | Diagonal Hessian methods | $O(N, K, B, P)$, $K > 1$ |
> | PackNet / LwF | At least one extra SFT-scale pass or an extra forward per step |
>
> Thus, for LLM-scale models, DSR offers **importance estimates at a fraction of the FLOPs** required by Fisher/Hessian-based or distillation-based baselines, while keeping the training loop as simple as vanilla SFT.
>
> **We have highlighted these comparisons in Section 5.1, and included the computing complexity comparisons in Appendix B.2 of the revised manuscript.**

---

> ### Author Response · Authors · 2025-11-21
> **Reply (3/3) to W3-W4, Q1-Q3**
>
> ### **W3. Extending to more experiments to validate generality.**
> Our research focuses on diverse reasoning capabilities, and commone reasoning tasks are math, code, logic, commonsense, which have been incorporated in our experiments. In the original manuscript, we evaluate the proposed DiFT on 2 and 3 reasoning tasks, demonstrating the consistent effectiveness. Indeed, more tasks can enhance the scalability ofthe DiFT, so we perform 4-task and 5-task experiments, results are as below:
> | Llama3-8B | GSM8k | xGLUE | LogiQA2 | CSQA | HotPotQA |
> | -- | -- | -- | -- | -- | -- |
> | mix-math-code-logic-csqa | 67.93 | 1.0058 | 31.55 | 77.07 | - |
> | +DiFT | **69.22** | **1.1342** | **33.21** | **77.31** | - |
> | mix-math-code-logic-csqa-hotpotqa | 62.62 | 0.9971 | 33.14 | 77.97 | 36.74 |
> | + DiFT | **62.68** | **1.0413** | **33.84** | **79.36** | **38.45** |
>
> Empirical results of 4-task and 5-task mix-up DiFT illustrate consistent improvements, demonstrating the effectiveness and more tasks potential of DiFT.
>
> **We have highlighted these contents in Section 5.2, and added the 4-task and 5-task experiments and explanations in the revised manuscript's Appendix F.5.**
>
> ### **W4. DSR sparsity changes with model size**
> DSR is a forward‑only, architecture‑agnostic statistic (row‑wise activation delta value) whose definition does not depend on model size or parameter count. Empirically, we already verify cross‑family generality and scaling up to 14B (Llama‑3, Mistral, Qwen‑2.5), with equal or larger gains at 14B (Appendix E, Table 10). For Mistral-7B, the DSR parameters are about 4.2% to the full-parameter, as for Qwen2.5-14B , DSR is about 3.9%, and DSR occupies 5.3% in Qwen2.5-3B. So from 3B to 14B, DiFT trains merely 4-6% of weights consistently, indicating DSR sparsity remains a small roughly scale-invariant constant, rather than growing with model size.
>
> ### **Q1. DSR across random seeds.**
> In the probing SFT statge, we fine-tune the base LLM with 1k math samples, then inference 5 groups of 50 math samples to obtain the DSR intersection as introduced in section 3.3. These groups of data samples share majority DSR parameters as shown in the following table:
> | overlap ratio (%) | sample-42 | sample-43 | sample-44 |
> | --- | --- | --- | --- |
> | sample-42 | 100 | 98.75 | 99.84 |
> | sample-43 | 98.75 | 100 | 99.32 |
> | sample-44 | 99.84 | 99.32 | 100 |
>
> We can see that all the math samples exhibit high consistency DSR parameters. We also construct another noisy data subset containing both math and code samples (90% math, 10% code), then calculate DSR parameters with the little noisy data on math-tuned model. We further compute the overlap ratio between the noisy DSR and normal DSR, results are as follows:
>  |   | sample-42 | sample-43 | sample-44 |
> | -- | -- | -- | -- |
> | overlap ratio (%) | 93.28 | 95.10 | 94.46 |
>
> We can see that a little noisy cannot affect the stability of the DSR computation to a large margin, demonstrating the consistency. However, the corruppted data subset indeed underperforms those pure single-task data as the former table, therefore, with pure data we can identify DSR within finetuned models.
>
> ### **Q2. Online/post-hoc DSR.**
> In practice, DSR can be made online, as it is essentially measuring **integrated row-wise update/gradient**, it shares a similar ideology with diagonal Fisher/importance estimates used in online continual-learning methods such as EWC and its online variants, which accumulate parameter-wise squared gradients over time to detect important weights.
>
> Analogously, one could maintain, during training, a running statistic per row (e.g., sum or EMA of (|\Delta W_k|^2) or (|g_k|^2) split by task), and periodically derive DSR-style masks from these online estimates. Those masks could then drive **on-the-fly conflict detection and adaptive freezing** of high-importance rows for previous tasks.
>
> In this paper, we focus on the simpler post-hoc variant (clearer analysis, no change to the training loop), and leave efficient online implementations as valuable future work.
>
> ### **Q3. How sensitive is DiFT to the hyperparameter controlling the top-C DSR threshold?**
> In Appendix D.3 (original manuscript's D.1), we tried 50, 100, 150, and 200 as different DSR numbers, and results illustrate that 100 is a better choice but others still work, demonstrating the robustness of the top-C DSR.
>
> **References:**
>
> [1] Overcoming catastrophic forgetting in neural networks. PNAS 2017.
>
> [2] Continual Learning Through Synaptic Intelligence. ICML 2017.
>
> [3] Memory Aware Synapses: Learning what (not) to forget. ECCV 2018.
>
> [4] Gradient Surgery for Multi-Task Learning. NeurIPS 2020.
>
> [5] Editing Models with Task Arithmetic. ICLR 2023.
>
> [6] Outlier Dimensions Encode Task-Specific Knowledge. EMNLP 2023.
>
> [7] Intrinsic Dimensionality Explains the Effectiveness of Language Model Fine-Tuning. ACL 2021.
>
> [8] Transformer Feed-Forward Layers Are Key-Value Memories. EMNLP 2021.

---

### Official Review · Reviewer_nbCX · 2025-11-03

**Soundness:** 3
**Presentation:** 4
**Contribution:** 3
**Rating:** 6
**Confidence:** 4

**Summary:**

This paper investigates the "task conflict" problem prevalent in multi-task (mixed or continual) supervised fine-tuning of LLMs, where fine-tuning on diverse reasoning datasets often leads to performance degradation in specific abilities. To address this, the paper first introduces an analysis method called "delta-scale row (DSR)," which identifies "exclusive parameters" critical to specific tasks by comparing the parameter activation differences between a base model and models fine-tuned on a single task. Building on this discovery, the authors propose the "Differential Fine-Tuning (DiFT)" framework: for mix-up SFT, only the union of the exclusive parameters for all relevant tasks is updated; for continual SFT, only the difference-set (parameters exclusive to the new task relative to old tasks) is updated. Experimental results on Llama3-8B, Mistral-7B, and Qwen2.5-14B demonstrate that DiFT consistently improves performance in the mix-up setting and effectively mitigates performance collapse in the continual setting.

**Strengths:**

1. The paper is well-motivated. Through solid preliminary experiments (e.g., Figure 1, Table 1), the authors clearly reproduce and quantify the 'conflict' phenomenon that arises in multi-task reasoning SFT, providing a strong impetus for their subsequent investigation.
2. The paper introduces the DSR diagnostic tool and the DiFT method. Instead of merely observing the superficial phenomenon that 'tasks conflict,' the authors delve deep into the model's internals to pinpoint the specific parameter subsets responsible for these conflicts. This "diagnose-then-treat" research paradigm is rigorous and well-founded.
3. The authors provide validation covering both 'mix-up' and 'continual' fine-tuning scenarios. Crucially, the ablation studies on 'Inverse DiFT' and 'Random DSR' convincingly demonstrate that the performance gains stem from correctly identifying critical parameters via DSR, rather than being a simple side effect of parameter reduction (i.e., regularization).

**Weaknesses:**

1.The paper lacks a robust theoretical foundation. While DSR is an interesting empirical discovery, the paper fails to provide a solid theoretical explanation for the underlying mechanism—why do LLMs tend to encode capabilities for different tasks into distinct parameter subsets?
2. The task scalability of the mix-up strategy is questionable. DiFT relies on updating the parameter union, but the experiments are limited to K=2 or 3 tasks. As K increases (e.g., in a practical scenario involving 20 tasks), the size of this union set could rapidly inflate, potentially approaching the full set of parameters and causing the DiFT strategy to degenerate into standard SFT. The paper lacks an analysis of DiFT's performance when K > 3.
3. The DSR method introduces significant workflow complexity. The framework requires an additional pre-computation stage: a separate "probe model" must be trained for each of the N single tasks before the main fine-tuning can commence. This requirement increases the complexity of the training pipeline.
4. The method's applicability to larger-scale models and different architectures is unproven. The experiments are confined to 7B-14B scale dense models, and it remains unknown if the approach holds for 30B+ models. The authors' discussion of MoE limitations (due to unstable routing) is reasonable, but this concession explicitly restricts the method's current applicability.

**Questions:**

1. DSR identification relies on a "probe model" trained with 1k samples. How stable is this DSR set? Would the identified set of "critical" parameters change significantly if identified using the full 20k dataset, or if the DSR analysis were re-run after the DiFT mix-up SFT? This questions whether DSR is a static property or one that evolves dynamically during training.
2. The mix-up SFT strategy relies on updating the parameter union. Have the authors investigated how the size of this union set scales as the number of tasks, K, increases (e.g., to K=5)? Is there a tipping point at which this union becomes so large that the DiFT strategy effectively degenerates into standard full-parameter SFT?
3. The continual SFT strategy avoids conflicts by freezing parameters from historical tasks. However, for the parameter intersection—those parameters critical to both old and new tasks—does this freezing strategy cause the model to "under-learn" the new task? Or, do the authors believe the benefit of mitigating forgetting always outweighs this risk of under-learning?
4. The authors state in the limitations that DSR is not directly applicable to MoE models due to unstable routing—a reasonable constraint. However, how do the authors conceptualize the relationship between DSR and the MoE architecture? Since MoE itself is a form of architectural parameter specialization, could the "exclusive parameters" identified by DSR be interpreted as "emergent experts"? Put differently, in an MoE model, is it hypothesized that a task's DSR would be primarily concentrated within its corresponding expert?
5. DSR successfully identifies which parameters are in conflict, but not fully why they conflict. Do the authors have a deeper hypothesis? For example, in Figure 7, the DSR pattern for csqa (commonsense) (rows j, k, l) appears distinctly different from the patterns for math, code, and logic (rows a-i). Does this imply that the root cause of the conflict is the model's attempt to encode fundamentally different types of information (e.g., symbolic manipulation vs. factual knowledge) within the same set of parameters (like mlp.gate_proj)?

---

> ### Author Response · Authors · 2025-11-21
> **Reply (1/3) to W1-W3**
>
> Thank you for your time and efforts! We really appreciate the constructive reviews, here are our responses to your raised concerns and questions.
>
> ### **W1. Lack a robust theoretical foundation for abilities encoded in different parameter subsets.**
> We agree that why LLMs tend to encode different task capabilities into distinct parameter subsets is an important and interesting phenomenon to be investigated, and that **there is not yet a solid theoretical explanation for this mechanism**. In this paper, our research goal is to identify and characterize the DSR phenomenon and demonstrate its practical implications for mitigating task conflicts happened in SFT. We did not claim to have a fully developed theory for *LLMs encode different abilities into discrete parameter subsets*. Rather, **we position DSR as a theoretically inspired and empirical discovery consistently observed across different models and tasks, and as a starting point for further theoretical work**. We have revised the paper to state this explicitly and avoid potential confusions.
>
> Though we do not aim to provide a complete theory for the *LLMs ability-subset* phenomenon, DSR is not purely descriptive. We offer several pieces of evidence that go beyond surface-level observations, including the **accumulated gradient-based derivations** in Section 3.2, the **DSR distribution variations analysis** in Section 3.3. To illustrate the differences among distinct tasks, we further statistic the overlap rate of task combinations as follows:
> | Comb. | math-code | math-logic | math-csqa | code-logic | logic-csqa |
> | -- | -- | -- | -- | -- | -- |
> | Overlap (%) | 43.2 | 31.1 | 20.7 | 31.1 | 20.7 |
>
> where math and code share nearly half parameters, and other combinations have 1/5 to 1/3 common DSR at most, indicating distinct reasoning tasks depend on different parameter subsets. In Appendix D.4, **single-task DiFT experiments** illustrate that full-SFT math is on par with DiFT math (61.64 vs 59.36), so does full-SFT and DiFT-code (1.1203 vs 1.1097), demonstrating that task abilities are encoded in corresponding DSR parameters.
>
> Although researchers always seek to understand the mechanism of LLMs, given the black-box nature and complexity of LLMs, many common circumstances, such as in-context learning and zero-shot prompting, still lack developed theoretical supports [1-6]. However, we can also conduct research on these topics, Task Vector aimed to steer the behaviors of pre-trained LLMs [3], Function Vector discovered in-context learning tasks related parameters. These works validated the task-parameter concurrence phenomena via experiments, our DSR conforms similar research paradigm, either.
>
> **We have highlighted that in Section 4.1 added details in the revised Appendix D.1.**
>
> ### **W2 The task scalability of the mix-up strategy is questionable.**
> We acknowledge that the current experiments focus on relatively small numbers of tasks (K=2 or 3), and thus cannot fully demonstrate scalability of DiFT on dozens of tasks. Now, we add 4-task and 5-task DiFT experiments, and results are as below:
> | Llama3-8B | GSM8k | xGLUE | LogiQA2 | CSQA | HotPotQA |
> | -- | -- | -- | -- | -- | -- |
> | mix-math-code-logic-csqa | 67.93 | 1.0058 | 31.55 | 77.07 | - |
> | +DiFT | **69.22** | **1.1342** | **33.21** | **77.31** | - |
> | mix-math-code-logic-csqa-hotpotqa | 62.62 | 0.9971 | 33.14 | 77.97 | 36.74 |
> | + DiFT | **62.68** | **1.0413** | **33.84** | **79.36** | **38.45** |
>
> Results of 4- and 5-task mix-up experiments illustrate consistent improvements, demonstrating the effectiveness and scalability of DiFT. There are additional complex reasoning, such as legal, and medical reasoning tasks, which can also be composed by math, code, logic, commonsense tasks. Given that, extensive reasoning tasks are actually decomposed into basic ones, and mix-up parameter union expands slowly, more details are in **our response to your Question 2**.
>
> **We have highlighted that in Section 5.2, added the experiments and explanations in the revised Appendix F.5.**
>
> ### **W3. DSR introduces extra complexity.**
> The "probe SFT" is indeed an extra step compared to vanilla SFT, however, such a step introduce quite limited resources: a 1k (1/20 than full) Llama3-8B SFT experiment consumes merely 1/20 time costs (6 minutes on 8xA100), and inference on the same serve needs less than 2 minutes. Therefore, the "probe" stage introduces 8 minutes, given the full SFT on the same device demands 120 minutes, which is 15 times of the "probe" step, the "probe" complexity is marginal.
>
> With the extra 1/15 cost, we can **produce an efficient SFT of merely 3%-5% trainable parameters, saving much time and computing than full SFT**. More importantly, our DiFT strategy provides **better interpretability** by locating task-specific DSR parameters, and achieve **superior diverse reasoning performance** through mitigating conflicts in SFT.
>
> **We have supplmented the details in the updated Appendix B.2 and B.3.**

---

> ### Author Response · Authors · 2025-11-21
> **Reply (2/3) to W4, Q1-Q2**
>
> ### **W4. Applicability to models scales and architectures.**
> In the initial manuscript, we verify the proposed DiFT on Llama‑3, Mistral, and Qwen‑2.5 across from to 14B. Due to that full SFT with 32B LLMs demands more GPUs, as we mentioned in the **Limitation section**, we have not enough computing resources to conduct for now. We promise to conduct that once getting enough resources.
>
> Even we have such resource limitations, to durther validate the scalability of our strategy, we conduct experiments on Qwen2.5-3B, and results are as follows:
> | Qwen2.5-3B | GSM8k | xGLUE | LogiQA2 |
> | -- | -- | -- | -- |
> | mix-math-code | 68.01 | 1.2102 | 35.69 |
> | +DiFT | **69.83** | **1.2213** | 35.94 |
> | mix-code-logic | 55.12 | 1.1997 | 36.90 |
> | +DiFT | 61.87 | **1.2375** | **37.10** |
> | seq-math-code | 60.05 | 1.1910 | 35.50 |
> | +DiFT | **61.79** | **1.1924** | 35.20 |
> | seq-math-logic | 62.93 | 0.9872 | 36.70 |
> | +DiFT | **64.37** | 0.9912 | **37.60** |
>
> We can see that DiFT can also work very well on Qwen2.5-3B on both mix-up and continual SFT settings. Despite the computing limitation, our experiments on 3B models, in conjunction with results from 7B to 14B models, consistently demonstrate the effectiveness of our method, suggesting its scalability with increasing model size (3B->7B->8B->14B). The DiFT strategy relies solely on DSR parameter selection, and larger models exhibit fundamentally similar DSR parameter nature. Furthermore, validating the consistent improvements of our method on models of this scale is particularly meaningful, as these model sizes are commonly employed in real-world deployments.
>
> **On MoE architectures and routing stability Regarding MoE models.** We indeed point out that unstable routing can complicate the direct application of our method in the Limiations. We would like to clarify additional 2 points:
>
> a. **Current limitation is practical, not conceptual.** Our concern with MoE is primarily about the instability of expert routing during training, more stable parameter subsets need more data to locate, which makes the notion of a stable task-specific parameter subset harder to define and track. This is a practical challenge, rather than a theoretical incompatibility between our method and MoE architectures.
>
> b. **A foundation for future MoE extensions.** We see our current dense-model study as a foundation. Understanding and validating the method in a simpler, more stable dense setting is a natural first step before tackling the additional complexity introduced by dynamic routing in MoE.
>
> **We have highlighted these in Section 4.1, and added more details to revised Appendix D.1 and F.6**
>
> ### **Q1. DSR identification relies on a 1k-trained "probe model".**
> We have fine-tuned the base LLM with 3 couples of 1k math samples and a full SFT with 20k math data, which are of no overlapped data, and use the same inference data to compute DSR of these finetuned LLMs. These math-finetuned LLMs share majority DSR parameters as shown in the following table:
> | overlap ratio (%) | math-1k-42 | math-1k-43 | math-1k-43 | math-20k |
> | --- | --- | --- | --- | --- |
> | math-1k-42 | 100 | 97.42 | 98.75 | 98.36 |
> | math-1k-43 | 97.42 | 100 | 98.07 | 97.92 |
> | math-1k-44 | 98.75 | 98.07 | 100 | 99.04 |
> | math-20k | 98.36 | 97.92 | 99.04 | 100 |
>
> We can observe that all the math training data exhibit high consistency DSR parameters, which is in line with the derivations in Section 3.2 and the empirical phenomena in Section 3.3.
>
> ### **Q2. How the size of the mix-up union set scales as the number of tasks increases?**
> As we introduced in the manuscript, different reasoning proficiencies may share some common DSR parameters like math-code (exhibited in Figure 2, Figures 7-10), therefore, the DSR union can not easily as the same scale as the full SFT. We notice that distinct tasks have different trainable parameter scales, we statistic the overlap rate of DSR parameters between any newly added task and historical tasks with the task number (K) increasing:
> | Task Num | Task | overlap rate (%) | param ratio (%) |
> | -------- | ------- | ---------------- | ------ |
> | K=1 | math | — | 3.7 % |
> | K=2 | code | — | 5.8 % |
> | K=3 | math | 54.3 | 7.2 % |
> | | code | 52.5 | |
> | | logic | 46.8 | |
> | K=4 | math | 63.8 | 8.6 % |
> | | code | 57.2 | |
> | | logic | 53.5 | |
> | | csqa | 63.4 | |
> | K=5 | math | 70.5 | 10.5 % |
> | | code | 67.2 | |
> | | logic | 68.9 | |
> | | csqa | 78.6 | |
> | | hotpoqa | 41.4 | |
>
> From this table, we can notice that with the task number increasing, each single task has more overlapped DSR parameters, thereby the growth of total trainable parameters slows down, showing a convergent trend. Such results indicate that the trainable parameter scale cannot easily as the same as full SFT, as extensive reasoning tasks can be decomposed into fundamental ones, and mix-up parameter union cannot expand very fast.
>
> **We have highlighted these in Section 5, and added these results in the updated manuscript's F.5.**

---

> ### Author Response · Authors · 2025-11-21
> **Reply (3/3) to Q3-Q5**
>
> ### **Q3. Would freezing parameter intersection strategy cause the model to "under-learn" the new task?**
> Empirical results of vanilla and DiFT continual SFT in Table 2, such as continual-math-code vanilla and DIFT (0.9902 v.s. 1.0557), show that freezing the intersected DSR can acheive better performance that full continual SFT, illustrating **the under-learning is slight**. And from the derivation of DSR in Section 3.2, if a subset of parameters is critical to both historical and current tasks, the magnitude of their updates have already been effectively captured by the DSR algorithm, **indicating that these parameters have been well optimized during prior learning phases. Therefore, these parameters can cause limited under-learn theoretically**.
>
> Nevertheless, the concern you raised is reasonable. We believe that mitigating reasoning conflicts and under-learning depend on specific scenario, our strategy support a trade-off between them: **we can introduce a mask ratio hyperparameter, to control the degree of under-learn and balance the target reasoning tasks**.
>
> ### **Q4. Conceptualize the relation between DSR and the MoE architecture.**
> We view DSR and MoE as complementary forms of specialization. In dense models, DSR reveals "soft experts": dispersed parameter subsets whose gradients dominate for a given task. MoE makes such specialization explicit via routed experts, and prior work shows experts naturally specialize in different input patterns or subtasks [8]. If routing were stable, we would indeed expect a task's DSR to concentrate in the experts most frequently selected for that task. However, **routing modules in practical sparse MoE training make such attribution brittle**, studying this correspondence is promising future work but orthogonal to our dense-model results.
>
> ### **Q5. Deeper hypothesis of DSR conflict.**
> During our investigation and analysis, we have a deeper guess that:
> > Math, code, and logic reasoning abilities mainly need to activate symbolic, algorithmic circuits, such as precise composition, intermediate-state tracking, and long-range dependencies. However, they may emphasize on a specific one or a combination, leading to the conflicts.
> > Commonsense reasoning and other similar tasks guide the LLM toward semantic, world-knowledge paths, like association, prototype matching, and heuristics, which demands totally different basic abilities.
>
> Given these functions coexist in the same MLP channels and attention heads, their gradients sometimes pull weights in incompatible directions [6]. DSR exposes where this happens, and we believe that understanding the exact circuits why is an exciting direction for future mechanistic work.
>
> **We have highlighted that in Section 4.1, and included this hypothesis at the end of Appendix D.1 of the updated manuscript.**
>
> **References:**
>
> [1] Overcoming catastrophic forgetting in neural networks. PNAS 2017.
>
> [2] Gradient Surgery for Multi-Task Learning. NeurIPS 2020.
>
> [3] Editing Models with Task Arithmetic. ICLR 2023.
>
> [4] Function Vectors in Large Language Models. ICLR 2024.
>
> [5] Intrinsic Dimensionality Explains the Effectiveness of Language Model Fine-Tuning. ACL 2021.
>
> [6] Transformer Feed-Forward Layers Are Key-Value Memories. EMNLP 2021.
>
> [7] Outlier Dimensions Encode Task-Specific Knowledge. EMNLP 2023.
>
> [8] Scaling and evaluating sparse autoencoders. arXiv:2406.04093.
>
> [9] Switch Transformers: Scaling to Trillion Parameter Models with Simple and Efficient Sparsity. JMLR 2022.

---

### Meta-Review · Area_Chair_BdYo · 2026-01-09

**Summary:**

This paper investigates task conflicts in multi-task and continual supervised fine-tuning of large language models and proposes Differential Fine-Tuning (DiFT), a principled framework that selectively updates task-relevant parameter subsets identified via Delta-Scale Row (DSR) analysis. Reviewers agreed that the problem is important and timely, and that the paper goes beyond documenting interference phenomena by providing an interpretable, diagnose-then-treat approach. Extensive experiments across multiple models (Llama3, Mistral, Qwen) and diverse reasoning tasks demonstrate consistent gains in both mix-up and continual fine-tuning settings. I recommend acceptance.

**Reviewer Concerns:**

Reviewers raised concerns regarding the lack of a complete theoretical explanation for why task-specific parameter subsets emerge, scalability to larger numbers of tasks and model sizes, and the need for stronger causal validation and continual-learning baselines. These concerns were largely addressed in the rebuttal through additional theoretical grounding (linking DSR to gradient accumulation and Fisher information), expanded experiments on 4–5 task mixtures and smaller/larger model scales, cross-task “row transplant” causal analyses, and comparisons with classical continual-learning baselines such as EWC and L2-SP.

**Reviewer Scores:**

Reviewer scores ranged from marginal to clear accept, with multiple reviewers rating the paper above the acceptance threshold and others indicating they would not object to acceptance. Importantly, no reviewer maintained a strong objection after the rebuttal.

---

### Decision · Program_Chairs · 2026-01-26

Accept (Poster)